



# A Framework for Advancing Streamflow and Water Allocation Forecasts in the Elqui Valley, Chile

Justin Delorit[1], Edmundo Cristian Gonzalez Ortuya[2], Paul Block[1]

[1]Department of Civil and Environmental Engineering, University of Wisconsin-Madison, Madison, 53706, United States
[2]Department of Industrial and Civil Engineering, University of La Serena, La Serena, 1700000, Chile

*Correspondence to*: Justin Delorit (delorit@wisc.edu)

**Abstract.**

In many semi-arid regions, agriculture, energy, municipal, and environmental demands often stress available water supplies. Such is the case in the Elqui River valley of northern Chile, which draws on a limited capacity reservoir and annually variable snowmelt. With infrastructure investments often deferred or delayed, water managers are forced to address demand-based allocation strategies, particularly challenging in dry years. This is often realized through a reduction in the volume associated with each water right, applied across all water rights holders. Skillful season-ahead streamflow forecasts have the potential to inform managers with an indication of likely future conditions upon which to set the annual water right volume and thereby guide reservoir allocations. This work evaluates season-ahead statistical prediction models of October-January (austral growing season) streamflow at multiple lead times associated with manager and user decision points, and link predictions with a simple reservoir allocation tool.

## 1 Introduction.

The sustainability of many water systems is challenged by current climate variability, and may come under additional stress with changes in future climate and user demands. Concerns over increasing water scarcity have prompted progressive governments, intuitions, water resource managers, and end-users to adopt a wide variety of conservation policies, typically targeting supply augmentation or demand reduction at the basin or jurisdictional boundary scale (Tanaka et al., 2006). These decisions, which are ideally informed by a variety of models, are inherently uncertain across time-scales, and produce numerous risks stemming from human activity and hydroclimatic variability/change (Narula and Lall, 2009). Advanced hydroclimatic information is often attractive to progressive water managers to support management and planning of water systems (Barsugli et al., 2012). At the seasonal scale, a skillful streamflow forecast may allow more efficient water allocation and predictable tradeoffs between flows for energy, irrigation, municipalities, environmental services, etc. Such forecasts often provide the ability to prepare for anticipated conditions, not simply react to existing conditions, potentially reducing climate-related risks and offering opportunities (Helmuth et al., 2007). This may be especially informative in years with





extreme conditions (floods, droughts.)  Further motiviation stems from evidence that addressing climate variability as part of water development is key for stabilizing and improving country economies (Brown and Lall, 2006).

While improvements in seasonal climate forecast skill and advocacy for integration into risk reduction strategies are well documented (Barnston et al., 1994; Block, 2011; Block et al., 2009; Dee et al., 2011; Hansen et al., 2004; Mason and Stephenson, 2008), demonstrated use of forecasts in current water allocation and policy strategies is limited (Barnston et al., 1994; Christensen et al., 2004; Hamlet et al., 2002; Sankarasubramanian et al., 2009; Stakhiv, 1998). This is partially attributable to the wide-spread use of static operational policies, which may be based on average streamflow or the drought of record, and established with minimal to no accounting of uncertainty, thus limiting water system flexibility (You and Cai, 2008). Effectively translating emerging climate information into hydrology to support adaptable water resources decision-making, and ultimately policy, warrants further study.

The water system in the semi-arid Elqui Valley in north-central Chile's IV[th] Region (Fig. 1) is contending with increasing levels of water stress and demand, coupled with insufficient investment in infrastructure, taxing its ability to sufficiently meet multiple water uses and maintain environmental quality.  The Valley footprint is relatively small (< 10,000 square kilometers), but boasts elevation changes ranging from sea level in the west to nearly 5,000 meters in the east along the Andes, in the span of less than 150 kilometers.  The Atacama Desert lies just to the north.  The Valley is fed from a retreating glacier to serve its 600,000 inhabitants, and is very narrow, with vineyards and plantations covering the floor and increasingly moving up the Valley sides; forty three percent of the region's surface land area is devoted to agricultural activities (Cepeda and Lopez-Cortes, 2004).  Agricultural exports, particularly grapes, fruits, and avocados, dominate the Valley's economy (Young et al., 2009), and are maintained by an extensive irrigation channel system latticing the Valley, which diverts water from the main Elqui River.  The Puclaro reservoir is the dominant storage facility in the Valley, with a holding capacity of 200 million cubic meters (Fig. 1.) The reservoir provides irrigation for about 21,000 hectares of the Elqui Valley, as well as small-scale hydropower (5.6 MW capacity) and being a popular tourist destination, particularly for sailing and windsurfing (Cepeda and Lopez-Cortes, 2004).





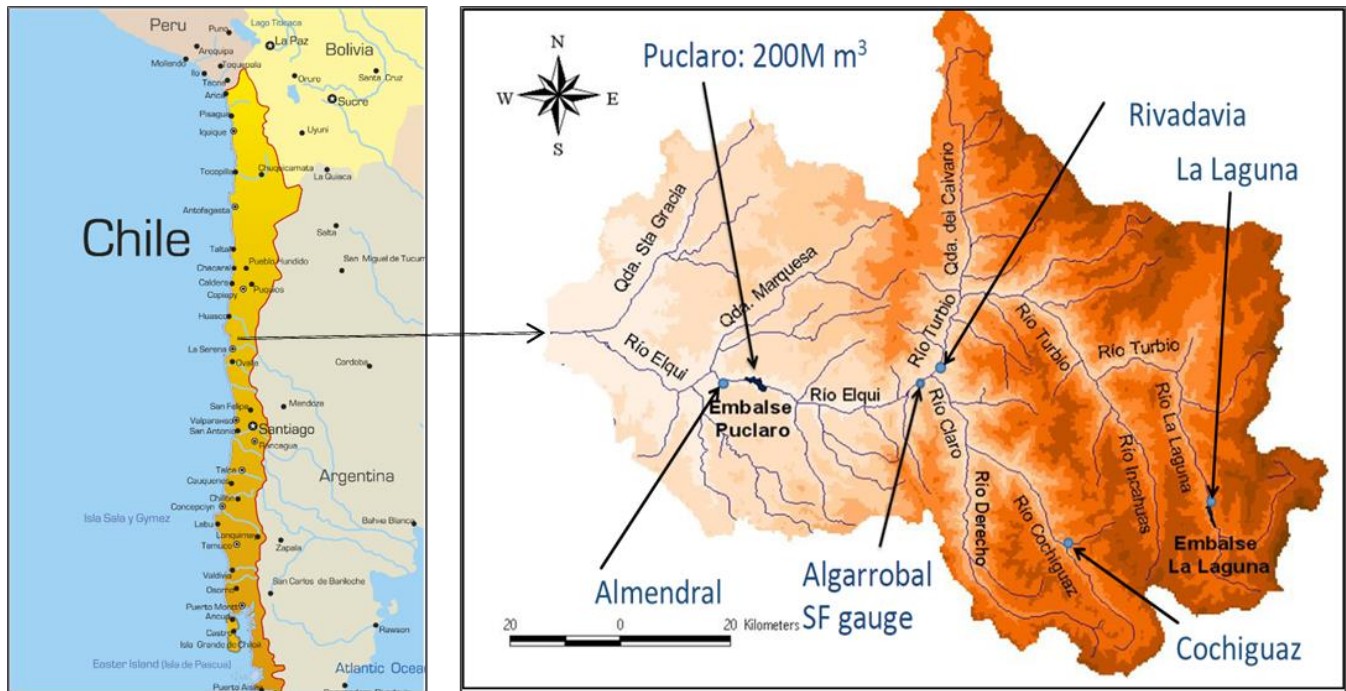

**Figure 1: Location of Elqui River Valley, Chile**

Chile uses a market-oriented approach to water allocation, guided by its Water Code of 1981 (G Donoso, 2006). The intent is to allow for optimal allocation and efficiency through a politically neutral mechanism via permanent trades or leasing (Olmstead, 2010; Wheeler et al., 2013). Rights are granted through the national water authority (Dirección General de Aguas, hereafter DGA), while supervision, reservoir management, and issuance of annual per right allocation is left to the privately-held, local water authority, Junta de Vigilancia del Rio Elqui (JVRE.) Water rights along the Elqui River are fully allocated, with 25,000 total rights valued at 1 liter per second each. In years with above normal precipitation and snowpack, this value can be attained, however near normal and below normal precipitation years typically require a reduction in per right allocation, on the order of 0.5 liters per second. Prolonged periods of drought (2009-2015) have resulted in allocations as low as 0.2 liters per second (JVRE, *personal communication*.) All water rights are of equal standing; no prioritization or junior/senior status exists. Thus, right holders above Puclaro are guaranteed equal per right allocations as their counterparts downstream; under the current framework, surplus supply cannot be allocated to users downstream of the reservoir once the annual per right allocation has been officially issued, to guarantee equality. Approximately 92% of water rights are held by farmers, with half of those held by a small minority engaged in large-scale viticulture. Municipalities and the mining industry share the balance of water rights. Meeting targets for renewable energy through hydropower, ecosystem services, specifically minimum instream flows, and reservoir storage are also important competing non-consumptive or non-water right holding priorities.



The decision framework driving water allocation and market activity in the Valley is complex and involves many actors. For the water year October to September, the local water authority initially projects the annual per right allocation in the preceding May and officially sets it in September. Water rights holders (users) thus have two decision points, May and September, to evaluate their allocation and weigh the need to supplement through market activity (trade or lease.) This setting serves as an

impetus for developing a framework to advance streamflow and water allocation forecasts at those decision points to better guide decision-making across the Valley.

## 1.1 Elqui Hydro-climate Characteristics.

The Elqui Valley is one of the most sensitive areas to water variability in all of South America, given its dryland ecosystem nature, susceptible to even small changes in the water cycle (Santibañez et al. 1992; N Kalthoff et al. 2006). The climate of

the region is affected by three major factors that lead to its semi-arid nature: the southeast Pacific anticyclone, the cold Humboldt current along the Pacific coast, and the eastern longitudinal barrier created by the Andes mountains (Kalthoff et al., 2002). The majority of precipitation is frontal in nature, falling in the austral winter (May-August, MJJA) as rain in the Valley and snow in the mountains; this leaves the remaining months extremely dry (Fig. 2;(Aceituno, Patricio, 1988). Annual rainfall totals approach 90mm on average and express a high degree of variability (Young et al., 2009). The El Niño Southern

Oscillation (ENSO) is well known to have a role in this variability, with positive precipitation anomalies during El Niño events, and below normal precipitation mostly associated with La Niña conditions (Fig. 3; (Aceituno, Patricio, 1988; Falvey and Garreaud, 2007; Garreaud et al., 2009; Montecinos and Aceituno, 2003). For Vicuña, a city located in approximately the center of the Valley, between 1950-2000, El Niño years produced average annual precipitation of 134mm, compared with 68 mm during La Niña years – a stark difference (Young et al., 2009).





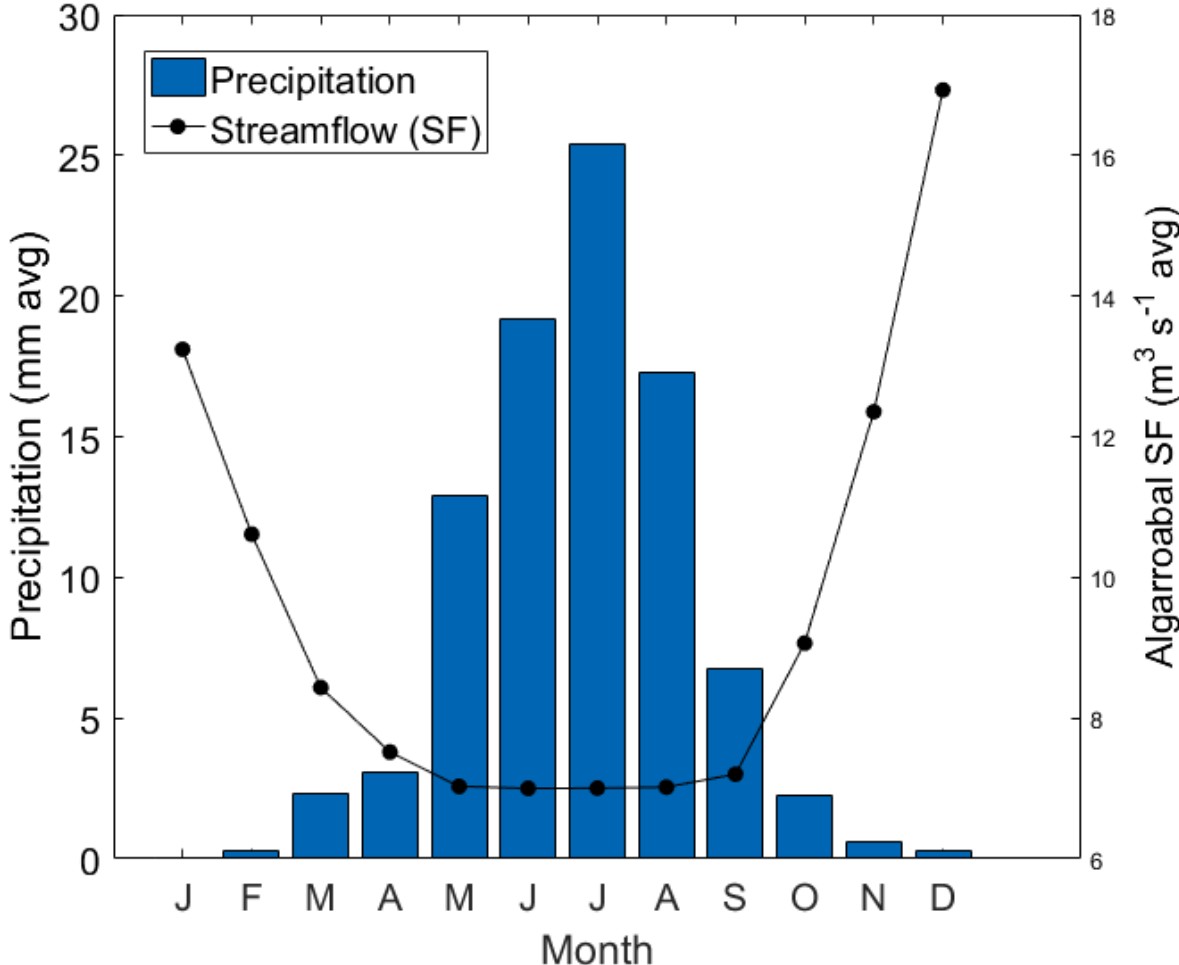

Figure 2: Annual cycle of average precipitation and streamflow (1950-2015)



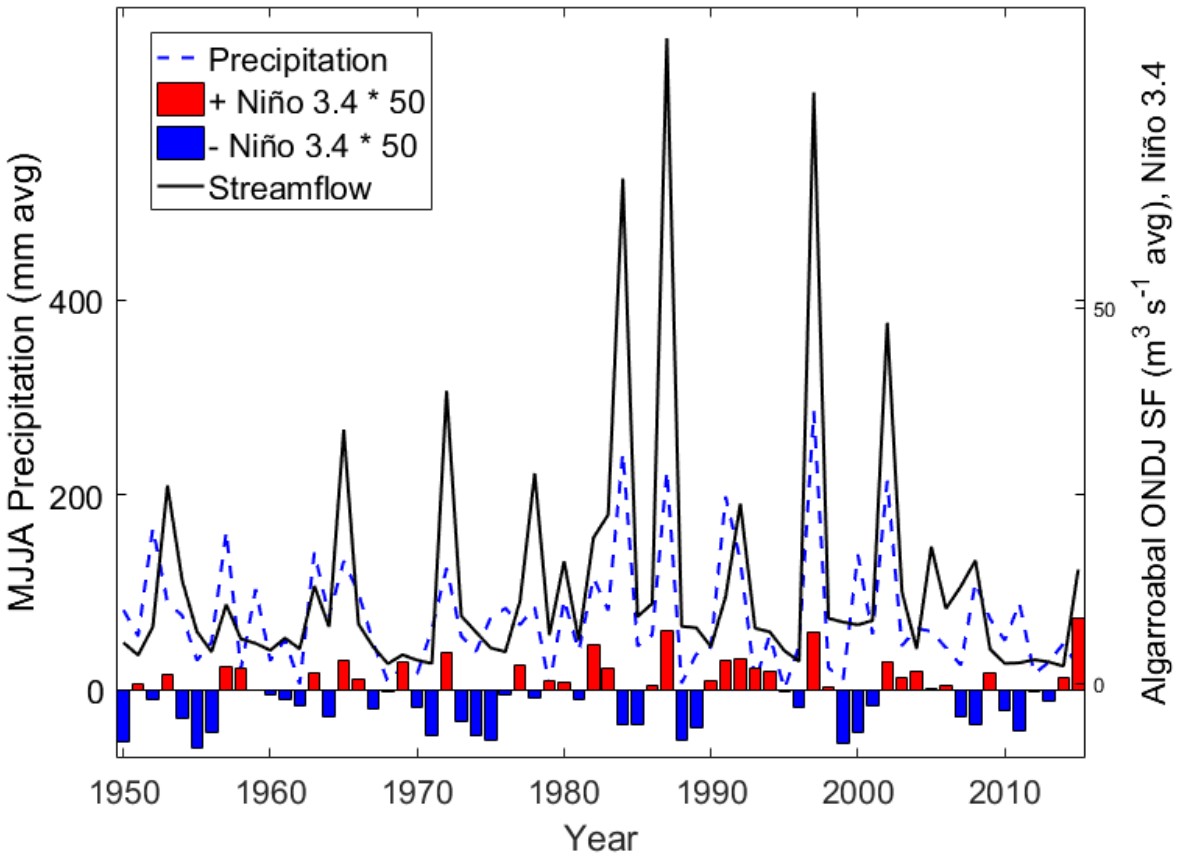

**Figure 3: Total annual precipitation (dashed), streamflow (solid) & May-August Niño 3.4 sea-surface temperature anomalies (bars)**

5   The Elqui River is predominantly fed through snowmelt over the October - January (ONDJ) season, dictating the agricultural calendar.  Historical rates of average streamflow over this season, however, indicate enormous interannual variability, ranging from 2.2 - 89 cubic meters per second at the Algarrobal station (Fig. 3; Santibañez et al. 1992), commonly considered as a surrogate for inflow to the Puclaro Reservoir (Fig. 1.)  Thus, a skillful streamflow forecast characterizing the ONDJ season has utility for providing advanced information to guide decisions in the Valley, particularly for dry conditions.

10  **2   Modeling Framework and Performance Metrics.**

Historically, water managers in the Elqui Valley have subjectively considered simple analog prediction models for ONDJ streamflow at Algarrobal, conditioned on the multivariate ENSO index (MEI), for allocation decisions and reservoir





operations, with limited success (JVRE, *personal communication*.) Previous efforts to evaluate hydro-climate forecast skill for the Elqui River have considered leads consistent with the current water rights forecast structure; a preliminary May allocation forecast and September allocation issuance (Robertson et al., 2014; Verbist et al., 2010). Roberston et al. (2014) report a significant increase in forecast skill, comparing September to May, but suggest further investigation to more fully

understand forecast skill with increasing lead time.

This recommendation is addressed by building a modeling framework to evaluate potential improvement in predicting ONDJ streamflow at multiple lead times, starting with a 1-month lead (September 1st) and increasing at monthly intervals (i.e. August 1st, July 1st, etc.) to May 1st, when the first water allocation forecast is preliminarily issued. Both statistical and dynamical

prediction approaches are explored. Subsequently, the ability to effectively predict water rights allocations is investigated by coupling streamflow predictions with a reservoir allocation model.

## 2.1 Statistical Streamflow Prediction Models

Statistical forecast methods rely on identification of spatiotemporal patterns in historical data (Chambers et al., 1971). Observations of streamflow at Algarrobal (monthly, 1948-present), valley-wide precipitation stations (daily, 1950-present),

and snow-water equivalent (daily, 1950-2009) are each readily available through the Chilean DGA. A suite of potential predictor variables are evaluated which have been shown to influence either streamflow or precipitation, including sea surface temperatures (SST), specifically in the Niño 1.2 and Niño 3.4 regions, sea level pressure (SLP), geopotential height, vector and meridional winds, local soil moisture, and the Multivariate ENSO Index (MEI), which combines several equatorial Pacific atmospheric and oceanic anomalies (Montecinos and Aceituno, 2003; Wolter and Timlin, 1993). These variables can illustrate

the mechanisms controlling moisture transport to the basin, and subsequent inter-annual variability in streamflow. For example, in the ten lowest ONDJ streamflow years (dry), vector winds follow a weak, dissociated pattern in the preceding season, which indicates that moisture transport from the Pacific Ocean is inefficient (Fig. 4(a.) In the ten highest ONDJ streamflow years (wet), vector winds are anomalously strong, and follow a coherent clockwise pattern off the coast of Chile, which suggests more efficient moisture transport is possible from the Pacific Ocean to the Elqui Valley (Fig. 4(b.)




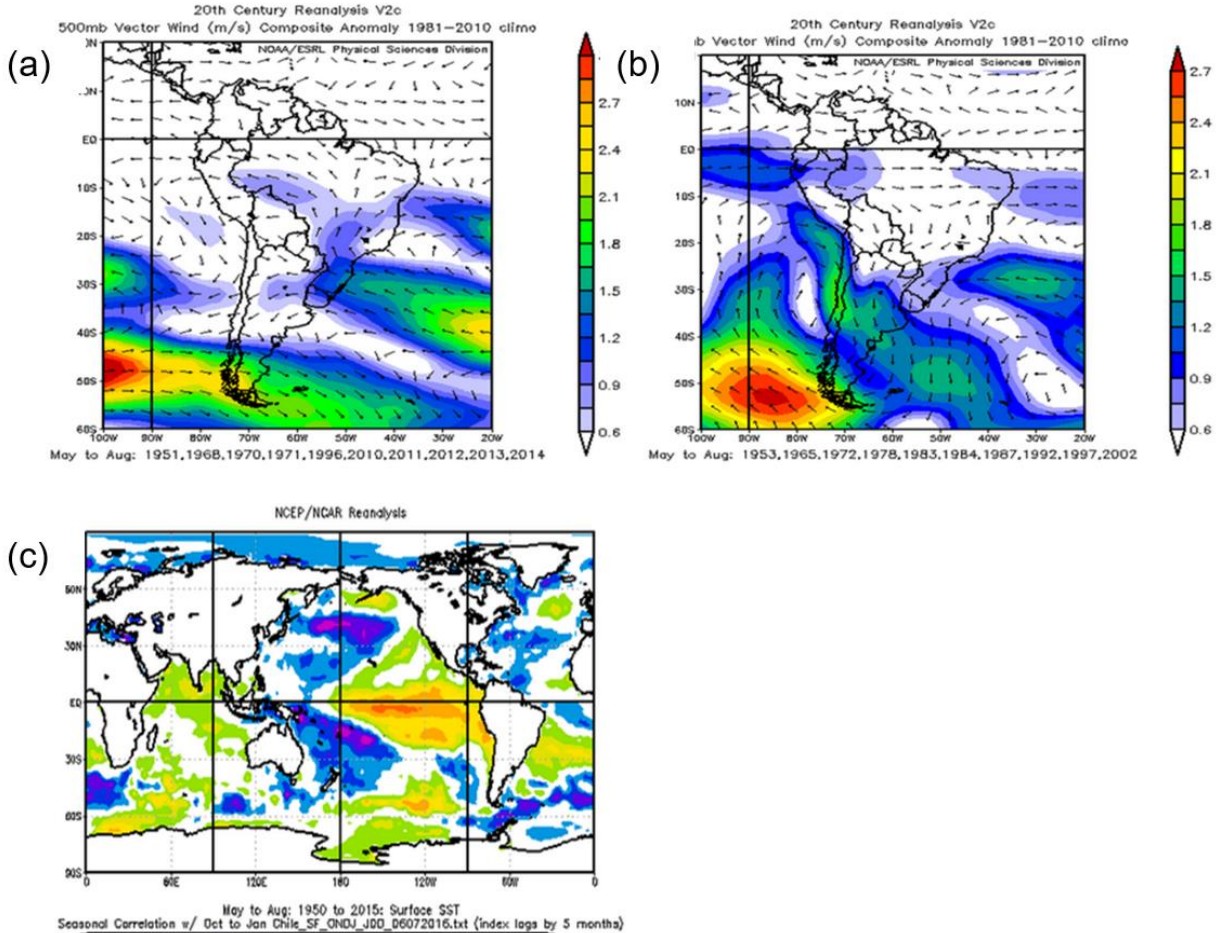

**Figure 4: (a) Composite May-August (MJJA) vector wind anomaly preceding ten lowest October-January (ONDJ) streamflow years, (b) same as (a) for ten highest ONDJ streamflow years, (c) correlation of MJJA sea-surface temperature anomaly with ONDJ streamflow (1950-2015)**

To identify potential predictors, each variable is correlated with ONDJ streamflow at lead times consistent with those discussed above (Fig. 5; not all variables shown.) Regions (gridded data sets) with statistically significant correlations in locations that have the potential to affect moisture transport (Table 1) are spatially averaged and retained for further evaluation. For example, the quintessential ENSO pattern in the equatorial Pacific Ocean is evident when correlating the entire ONDJ streamflow record

10    with SST anomalies in the preceding MJJA, which suggests ENSO, in general, plays some role in explaining streamflow variability within the Elqui Valley (Fig. 4(c.) SST, SLP, geopotential height, meridional and vector winds are obtained at a 2.5 x 2.5 degree grid resolution from the National Oceanic and Atmospheric Administration's Climate Diagnostics Center (NOAA-CDC), which are based upon the National Center for Environmental Prediction–National Center for Atmospheric





Research (NCEP–NCAR) reanalysis data, available from 1949 to the present (Saha et al., 2013). Soil moisture data is obtained from NOAA's Climate Prediction Center's (CPC) global monthly soil moisture dataset, at 0.5 x 0.5 degree grid resolution, which is available from 1948 to the present (Kalnay et al., 1996; Saha et al., 2013). MEI data is available from NOAA's Earth System Research Laboratory (ESRL) bimonthly as the first unrotated principal component of six spatially filtered variables in

5    the tropical Pacific (Wolter and Timlin, 1993, 1998).

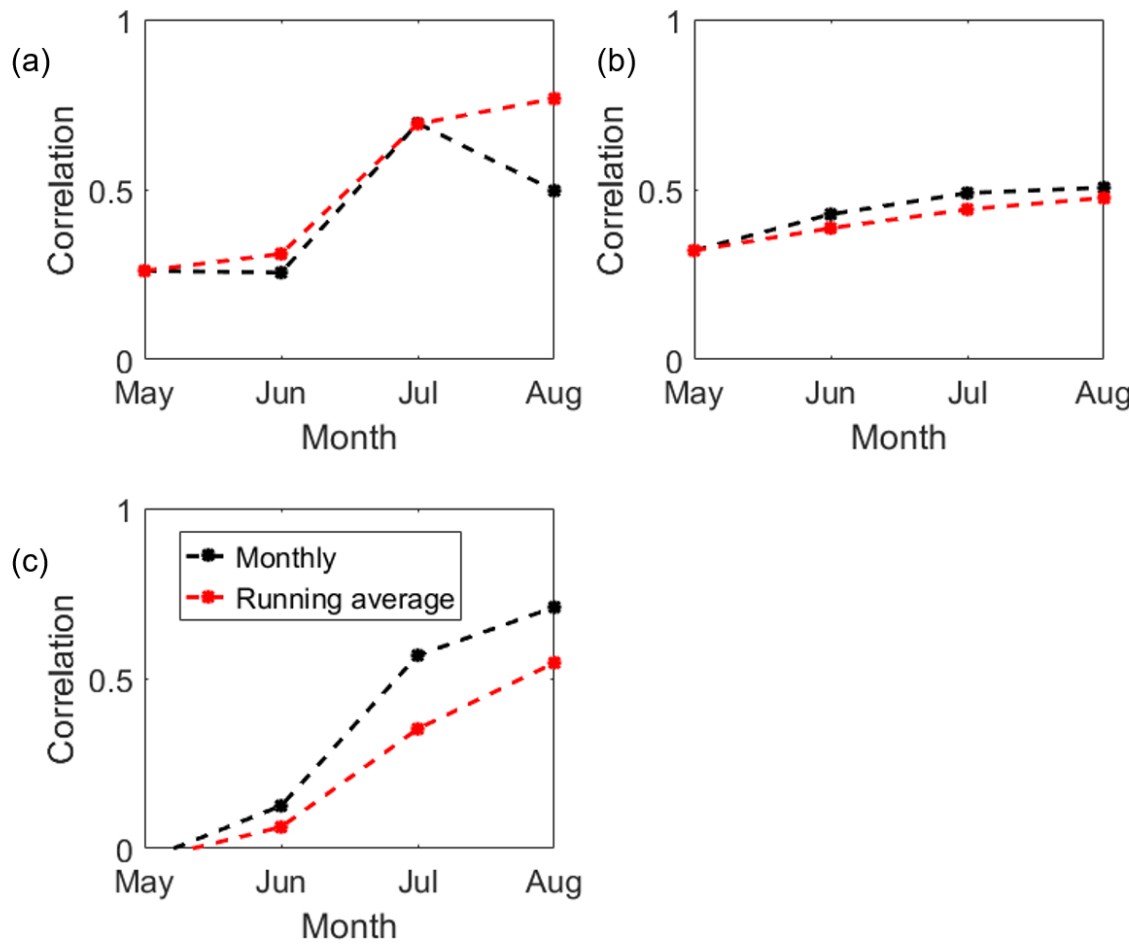

**Figure 5: Temporal correlations of October-January streamflow and potential predictors: (a) precipitation, (b) Niño 3.4 sea surface**
10    **temperatures, (c) soil moisture**



**Table 1: List of potential predictors (bold predictors retained for statistical model)**

| Potential Predictors | Location | MJJA Pearson's correlation with ONDJ Streamflow |
|---|---|---|
| *Local* | | |
| **Precipitation** | 18 Stations (Valley Wide) | +0.80 |
| **Soil Moisture** | 29°- 30° S, 70°-71° W | +0.55 |
| Snow-water equivalent | 1 Station (La Laguna) | +0.68 |
| *Global* | | |
| Geopotential Height (800 mb) | 45°- 60° S, 100°-120° W | +0.43 |
| Meridional Wind | 0°-5° S, 160°- 180° W | +0.37 |
| Multivariate ENSO Index | Tropical Pacific Anomaly | +0.35 |
| Sea Level Pressure | 60°- 70° S, 100°-120° W | +0.22 |
| Sea Surface Temperatures (Niño 1.2) | 0°-10° S, 80°- 90° W | +0.40 |
| **Sea Surface Temperatures (Niño 3.4)** | 5°N-5° S, 120°-180° W | +0.49 |
| Vector Wind | 20°- 60° S, 120°-180° W | +0.47 |

Principal component regression (PCR) (Lins, 1985) is commonly applied in forecasting to decompose space-time fields, which

reduces both dimensionality and multicollinearity of a set of variables. PCR is a two-step process, the first of which identifies modes of dataset variability iteratively, by identifying the direction which maximizes the variance explained in the data.  The resultant principal component (PC) is the sum of least squares distance between the factor direction and the predictor data. The second factor is applied in the direction which maximizes dispersion in the dimension of next greatest variability to form the second PC, and so forth.  All PCs are orthogonal.  The result is a set of PCs representing the variance in the predictors,

with PCs ordered by the amount of variance explained. PCs with eigenvalues greater than one are retained, following Kaiser's rule; (Zwick and Velicer, 1986). The second step of PCR is multiple linear regression, using the PCs retained as predictors as shown by Eq. (1):

$$\hat{y}_t = \beta_0 + \beta_1 x_1 + \beta_2 x_2 + \cdots + \beta_n x_n, \ for \ t = 1 \ to \ N \tag{1}$$

$$\varepsilon_t = \hat{y}_t - y_t \tag{2}$$

where $\hat{y}_t$ is the predicted value of ONDJ streamflow in year $t$, $x_n$, . . ., $x_n$ are the PCs retained as predictor variables, $\varepsilon_t$ is an error coefficient calculated as the difference of predicted and observed values of streamflow, as shown by Eq. (2), and $\beta_o$,. .

.,$\beta_n$ are the fitted regression coefficients. A leave-one-out cross validated hindcast is undertaken to produce a deterministic prediction of expected streamflow for each year (1950-2015) (Block and Rajagopalan, 2007).  A prediction distribution is generated using prediction errors ($\varepsilon_t$) from the hindcast fit to a normal distribution with a mean of zero, and added to the



deterministic hindcast prediction. In this work, the median and upper 80[th] percentile hindcasted flows from the ranked outputs are analyzed. The 80[th] percentile streamflow time series is used as a conservative estimate of streamflow to simulate potential risk aversion on the part of a reservoir manager. Hereafter the statistical principal component regression approach is referred to as Stat-PCR.

As previously mentioned, ENSO influences Elqui River streamflow variability. The strength of an El Niño or La Niña event relates to the degree of SST deviations from the long-term mean; using the Niño 3.4 Index, NOAA has established weak (+/- 0.25° C), moderate (+/- 0.75° C), and strong (+/- 1.0° C) categorical thresholds as a means of describing ENSO phase and strength (2016). Recent research has illustrated a potential relationship between streamflow and ENSO phase and strength (Zimmerman et al., 2016). In a separate statistical approach, a streamflow prediction model based on ENSO phase and strength (Stat-P&S) is developed to provide categorical predictions of ONDJ streamflow. The Stat-P&S approach utilizes Niño 3.4 Index values, prior to the ONDJ season of interest, to provide a categorical streamflow prediction. To qualify for prediction using Stat-P&S, at least one month during a selected Niño 3.4 Index window must be at least moderate in strength for a given phase, $\geq$ +0.75°C (El Niño) or $\leq$ -0.75°C (La Niña.) Years satisfying this criterion are categorically predicted as Above Normal (A; highest 33% of long-term streamflow observations) or Below Normal (B; lowest 33% of long-term streamflow observations) ONDJ streamflow, respectively. Window selection determines hindcast date, and may fall prior to or during a phenomenon known as the Spring Barrier, when SSTs in equatorial Pacific generally reset, losing predictive strength (Webster and Hoyos, 2010). However, the effects of moderate and strong ENSO events have some tendency to persist (Balmaseda et al., 1995). When values from the Niño 3.4 Index fail to exceed +/- 0.5°C, ONDJ streamflow is predicted to fall into the Normal (N; middle 33% of long-term streamflow observations) category. For years where the Niño 3.4 Index values are (+0.5°C, +0.75°C) or (-0.5°C, -0.75°C), the Stat-P&S model does not issue a forecast.

## 2.2 Dynamical Streamflow Prediction Model.

General Circulation Models (GCM) are physically-based, three dimensional representations of gridded atmospheric, oceanic and land surface processes, with typical spatial resolutions of 250 - 600 km (Giorgi, 1990). GCMs have proven skillful in prediction of large scale physical processes, such as SSTs and pressure systems, however, their relatively coarse resolution often limits predictive ability for smaller scale weather and climate phenomena, including precipitation (Bosilovich et al., 2008). Furthermore, outputs from each GCM are unique, and based on individualized parameterization schemes, initial conditions, data assimilation processes, etc. Considering the National Multi-model Ensemble (NMME; (2012) suite of models, (Verbist et al., 2010) demonstrate skillful prediction of North Central Chile precipitation based on equatorial Pacific SSTs in the ENSO region using NOAA's National Centers for Environmental Protection's (NCEP) Climate Forecast System Version 2 GCM, available 1982 – present (CFSv2; (Saha et al., 2013). Considering both the findings of Verbist et al. (2010), and a strong Pearson's correlation coefficient between observed ONDJ streamflow and MJJA precipitation in the Elqui Valley





(0.80), both precipitation and SSTs outputs from CFSv2 are retained for further evaluation. Specifically, the mean value of the 40-member ensemble of outputs for gridded precipitation (29˚- 30˚S, 70˚-71˚W) and the Niño 1.2 and 3.4 indices at leads between January 1$^{st}$ and May 1$^{st}$ are obtained and independently corrected using a statistical quantile mapping approach based on the cumulative distribution functions of both predicted and observed data (Maraun, 2013). For each lead, predicted values

5    are replaced with values from the observed distribution, based on matching probabilities (Fig. 6; not all variables shown.) The same PCR framework as in the Stat-PCR approach is applied using GCM corrected precipitation and SSTs to predict ONDJ streamflow, referred to as the Stat-Dyn model. For occurrences when local variables (e.g. precipitation, snow water equivalent and soil moisture) hold predictive strength, they are also added to the model.



**Figure 6: (a) Quantile mapping of predicted and observed NOAA NCEP CFSv2 Niño 3.4 sea surface temperature (SST) data, (b) observed, predicted and statistically corrected NOAA NCEP CFSv2 Niño 3.4 SST data**



**2.3 Allocation Forecast Model.**

Coupling the streamflow prediction models to a simple reservoir balance model, allocation and storage outcomes are hindcast for the period of record (1950 – 2015), which allows for evaluation of a per-right allocation, as issued annually by the JVRE. As previously mentioned, if allocations are reduced to less than the defined maximum of 1 liter per second, all rights are
reduced equivalently across rights holders, per Chile's water code. The Puclaro operating rules adopted here focus on the end of water year (February 1st) target reservoir volume, set at 100 million cubic meters (50% capacity), which is consistent with current management practices for Puclaro Reservoir. To account for annual deviation from the end of water year storage target, allocation for ONDJ in year $i+1$ is adjusted by the difference between end of water year storage and the target in year $i$. Allocations may be larger if end of year storage exceeds target storage, or smaller if there is a shortfall in end of year storage,
as shown by Eq. (3), where

$$A_{i+1,ONDJ_{prediction}} = \frac{Q_{i+1_{prediction}}}{\frac{WR_u}{WR_D}+1} - \left(100Mm^3 - S_{i,Feb_{adjusted}}\right) \tag{3}$$

$A_{i+1,ONDJ_{prediction}}$ is the predicted allocation for ONDJ in year $i+1$. $Q_{i+1_{prediction}}$ is the prediction of inflow in year $i+1$, with
streamflow predictions for the non-ONDJ months constructed by regressing median ONDJ streamflow predictions onto February – September streamflow observations to produce predicted February – September streamflow. $WR_u$ and $WR_D$ are the number of water rights upstream and downstream of Puclaro, respectively, and $S_{i,Feb_{adjusted}}$ is the previous end of water year adjusted storage volume, as shown by Eq. (4), where

$$S_{i,Feb_{adjusted}} = S_{i,Sep_{prediction}} - (A_{i,ONDJ_{prediction}} - A_{i,ONDJ_{observation}}) \tag{4}$$

$S_{i,Sep_{prediction}}$ is the predicted storage at the time of ONDJ allocation issuance in year $i$, and $A_{i,ONDJ_{prediction}}$ and $A_{i,ONDJ_{observation}}$ are the forecast-based and observed allocation values in year $i$. This adjusted volume accounts for storage deficit or surplus resulting from forecast-based allocations (forecasts will not be perfect, i.e. not match observations), and
allows for adjustment of allocation in the following year to either replenish the reservoir or provide additional allocation, respectively.

Annual per water right allocations based on forecasts of September 1st reservoir volume, probabilistic inflow predictions, and end-of-water-year target reservoir volumes, are reported as a probability of falling into three allocation categories: "Moderate"
($\geq$ 0.5 Liters per second), "Severe" (0.5 Liters per second – 0.25 Liters per second), and "Extreme" (<0.25 Liters per second.) The selected categories are consistent with those used by the U.S. Drought Monitor to describe similar ranges of industrial,





social and environmental impacts expected due to reduced access to water resources (Svoboda et al., 2002). Numerical thresholds assigned to the categorical boundaries align approximately with tercile values from the cumulative distribution of allocations derived from observed inflow and storage data, using the same reservoir operating rules as forecast-based allocations. Further, the breaks in categories closely follow decisions made by JVRE: a water right value of 0.5 liters per second is not uncommon and approximately represents the lower bound in normal years (Hearne and Easter, 1995); during the most recent severe drought (2009-2014) water right values of 0.2 liters per second were common (D. Betancourt, *personal communication.*)

### 2.4 Performance Metrics.

The performance of each modeling approach is assessed deterministically (Pearson's correlation coefficient) and with a variety of categorical metrics to assess model skill in the prediction of specific categories, as opposed to specific quantity or pattern (Regonda et al., 2006; Souza Filho and Lall, 2003). Two sets of categories are evaluated, as previously defined. The first is for streamflow hindcast prediction, with Above- (A), Near- (N), and Below-Normal (B) categories (ranges) based on a climatological distribution of observed ONDJ streamflow, each containing 33% of observations. The second is for per water right allocation hindcast prediction, applying the Moderate, Severe, and Extreme categories, as previous defined and contingent on reservoir storage and forecast inflow. Categorical outputs are illustrated with contingency tables, comparing predicted versus observed categorical occurrences. Perfect model skill occurs when predicted conditions match or 'hit' observed conditions. Equation (5) illustrates the 'Hit Score' summary metric, as applied by (Barnston, 1992), which describes the categorical performance of the entire forecast in comparison to observations, where

$$Hit\ Score = \frac{\sum(Hits_A, Hits_N, Hits_B)}{n} \times 100\% \qquad (5)$$

$\sum(Hits_A, Hits_N, Hits_B)$ is the sum of the count of years predicted correctly in each category, while $n$ is the total number of years in the record. Individual categorical Hit Scores describe under which flow conditions the model is most skillfull, and is the count of years predicted correctly in a category divided by the number of years observed in the same category. A 'Miss' results when the predicted value does not fall within the observed category. An 'Extreme Miss' constitutes a categorical prediction missing an observation by two categories (model predicts Above-normal while Below-normal is observed or vice-versa.) The 'Extreme Miss Score', as shown by Eq. (6), is the fraction of the sum of misses Above-normal is predicted but Below-normal is observed ($miss_{A/B}$) plus the sum of misses Below-normal is predicted but Above-normal is observed ($miss_{B/A}$) and the total number of hindcast years, $n$,

$$Extreme\ Miss\ Score = \frac{\sum miss_{A|B} + \sum miss_{B|A}}{n} \times 100\% \qquad (6)$$



Ranked Probability Skill Score (RPSS) is a categorical measure of an ensemble prediction of each modeling approach compared to a reference forecast, in this case climatology (SAUNDERS and FLETCHER, 2004). To calculate RPSS, the Ranked Probability Score (RPS), as shown by Eq. (7) must first be calculated for each simulation.

$$RPS = \frac{1}{N-1}\sum_{n=1}^{N}[\sum_{c=1}^{n}p_c - \sum_{c=1}^{n}o_c \ ]^2 \tag{7}$$

The RPS is a measure of square differences in the cumulative probability of a multi-categorical hindcast ensemble, where $N$ is the number of hindcast categories, $p_c$ is the probability of the predicted value in category $c$, and $o_c$ is a binary indicator with a value of one if the observation falls in category $c$, or a value of zero otherwise. The RPS ranges from 0 to 1, increasing for

10    predictions farther from the observed outcome.

The RPSS utilizes RPS, and ranges from -∞ to 1; values between 0-1 indicate greater skill than simply using climatology (i.e. basing prediction on long-term averages), while RPSS values less than zero indicate predictions are inferior to climatology. An RPSS value is generated for each of year of the hindcast using Eq. (8); the median RPSS value is reported.

$$RPSS = \frac{\overline{RPS} - \overline{RPS}_{reference}}{0 - \overline{RPS}_{reference}} = 1 - \frac{\overline{RPS}}{\overline{RPS}_{reference}} \tag{8}$$

## 3 Model Performance.

### 3.1 Statistical and Dynamical Streamflow Prediction Models.

20    For each streamflow modeling hindcast assessment (Stat-PCR: 1950 – 2015; Stat-Dyn: 1982 – present), a unique set of predictors and principal components are selected and evaluated with the categorical performance metrics (Pearson's correlation coefficient, 'Hit Score', 'Extreme Miss Score', and RPSS; Table 2.) Not surprisingly, as forecast lead increases, both Hit Score and RPSS decrease, while Extreme Miss Score increases, as less information regarding the MJJA rainy season is available, which is consistent with decreased correlations between ONDJ streamflow and predictors (Fig. 5.)




**Table 2: Stat-PCR and Stat-Dyn forecast model performance metrics**

| Forecast | | Retained Predictors | | | PC1 | PC2 | Pearson's Correlation Coefficient | Hit Score | Extreme Miss Score | RPSS |
|---|---|---|---|---|---|---|---|---|---|---|
| Statistical Approach (Stat-PCR) | Sep 1st | Aug SM | JA Prcp | Aug 3.4 | 89% | - | 0.88 | 53% | 11% | 0.31 |
| | Aug 1st | Jul SM | JJ Prcp | Jul 3.4 | 63% | 24% | 0.63 | 50% | 12% | 0.02 |
| | Jul 1st | Jun SM | MJ Prcp | Jun 3.4 | 44% | 38% | 0.49 | 31% | 24% | -0.39 |
| Dynamical Approach (Stat-Dyn) | Jun 1st | JJA 1.2 | JJA Prcp | - | 65% | 35% | 0.45 | 26% | 50% | -0.32 |
| | May 1st | JJA 3.4 | JJA Prcp | - | 58% | 42% | 0.41 | 21% | 53% | -0.41 |
| | Jan 1st | JJA 3.4 | | - | - | - | 0.38 | 20% | 57% | -0.76 |

For the Stat-PCR set of models, the predictors for each lead-time follow a similar pattern, utilizing soil moisture and SST from the month prior, and precipitation for the two months prior to the forecast date (e.g. September 1st forecast uses August soil moisture and SST, and July-August precipitation.)  The September 1st lead is promising, however for longer leads this relationship does not necessarily hold.  An August 1st lead is approximately equivalent to using climatology, and by July 1st it is worse.  For the Stat-Dyn modeling approach, using the mean of CFSv2 ensemble forecasts for MJJA precipitation, Niño 3.4

and 1.2 SSTs at Jun 1st, May 1st and January 1st lead times, produces low Hit, high Extreme Miss and negative RPSS scores (Table 2), affirming the challenges of predicting through the Spring Barrier.

The first principal component of the Stat-PCR September 1st forecast is highly correlated with SST in the Niño 3.4 region (0.88), which confirms that streamflow and therefore precipitation in the Elqui Valley are at least partially characterized by

anomalous changes in SSTs.  From a categorical perspective, the statistical model is most skillful in predicting Above-Normal streamflow years (Hit Score: 82%; Table 3); categorical outcomes for Near- and Below-Normal streamflow years were less successful (Hit Scores: 36% and 64%, respectively.)  The large disparity between Above-, Near-, and Below-Normal categorical outcomes may be explained by evaluating cross-validated, global spatial correlation maps (1° x 1°) of ONDJ streamflow with the MJJA MEI, following Zimmerman et. al (2016.)  The spatial correlation plots (1950 – 2015; Fig. 7)

illustrate that years with positive MEI generally correspond with El Niño events and Above-Normal streamflow conditions, while years with negative MEI generally correspond with La Niña events and Below-Normal conditions.  This produces a strong positive correlation (0.65) between streamflow and SST in the Niño 3.4 region during years with positive MEI, and a moderate positive correlation (0.29) during years with negative MEI in the equatorial Pacific Ocean, but slightly outside the common ENSO index regions.  Correlation mapping between all years and streamflow produces a moderate correlation (0.35)

in the common ENSO region, suggesting that El Niño years likely dominate this relationship.  However, ENSO is non-linear,





and the amount of moisture transported to the basin during El Niño or La Niña years will vary dependent upon strength (Meehl et al., 2001), and other factors, as previously discussed and illustrated in Fig. 4.

**Table 3: September 1st Stat-PCR model categorical streamflow results: observed vs. forecast**

|  |  | Forecast – September | | |
|---|---|---|---|---|
|  |  | B | N | A |
| Observed | B | 14 | 3 | 5 |
|  | N | 6 | 8 | 8 |
|  | A | 2 | 2 | 18 |

*Below-normal (B) Near-Normal (N) Above-Normal (A)*

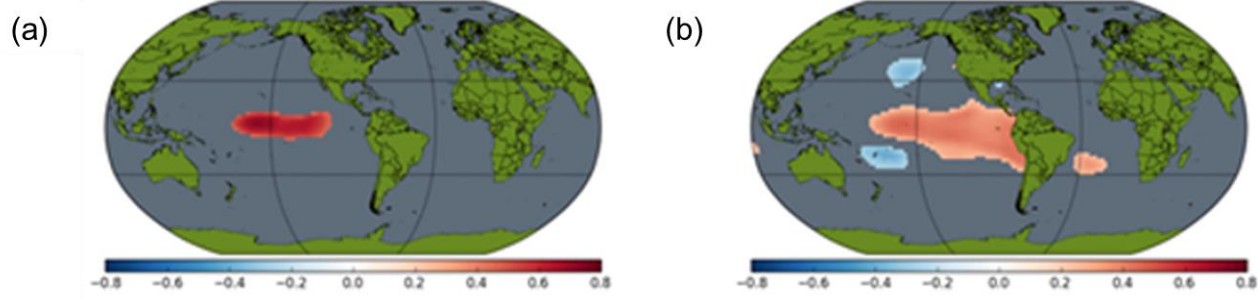

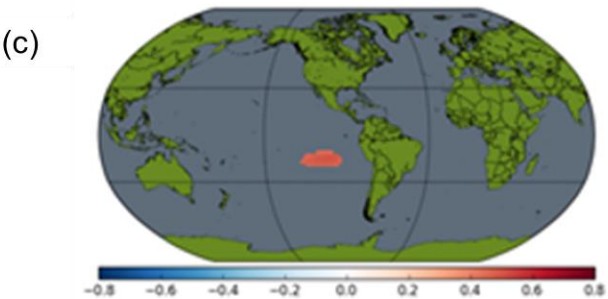

**Figure 7: May-August global Multivariate ENSO Index (MEI) correlated with October-January streamflow at Algarrobal for: (a) positive MEI years, (b) all MEI years, (c) negative MEI years**



### 3.2 ENSO Phase and Strength Streamflow Prediction Models.

To evaluate ENSO phase-specific models, the Stat-P&S approach is adopted. While several forecast leads and Niño 3.4 index windows were evaluated, the Stat-P&S model performs best for a May 1st forecast, when SSTs in the Niño 3.4 region are at

least moderate in strength for a given phase [≥ +0.75°C (El Niño) or ≤ -0.75°C (La Niña)], or relatively neutral [within +/-0.5°C departure from the long-term mean], for at least one month during January-April (JFMA; Table 4.) For 1950 – 2015, 60% of years qualify, triggering the May 1st Stat-P&S categorical prediction model. For moderate conditions (positive and negative), this produces categorical Hit Scores of 75% for Above-normal (El Niño) and 58% for Below-normal (La Niña.) For moderate La Niña only conditions, 7 of the 10 lowest ONDJ streamflow years on record are captured. The remaining three

years of lowest ONDJ streamflow (1969, 1995, 2010) are predicted as Above-normal by the Stat-P&S model due to JFMA Niño 3.4 SSTs > 1.0°C (strong El Niño conditions.)

**Table 4: Stat-P&S model categorical streamflow results: observed vs. forecast**

|  |  | Forecast – May | | | |
|  |  | B | N | A | DNF |
| Observed | B | 7 | 2 | 3 | 27 |
|  | N | 6 | 3 | 6 | |
|  | A | 3 | 0 | 9 | |

*Model does not forecast (DNF)*

### 3.3 Coupled Statistical Prediction Models.

The Stat-P&S and Stat-PCR models each provide skillful forecasts, at different leads. While Stat-P&S performs best for a May 1st forecast lead, particularly for predicting high and low ONDJ streamflow, forecasts are issued only categorically; deterministic predictions from the Stat-PCR and Stat-Dyn models at this lead are relatively weak. That is, the Stat-P&S model relinquishes forecast determinism and in turn increases forecast lead in comparison to the Stat-PCR and Stat-Dyn approaches.

The Stat-P&S approach model is also hindered in only being triggered in 60% of the period of record. The other 40% of years occur when Niño 3.4 SSTs for at least one month during JFMA are (+0.5°C, +0.75°C) or (-0.5°C, -0.75°C.) In contrast, while the Stat-PCR approach provides deterministic forecasts of ONDJ streamflow, it is only skillful at a September 1st forecast lead, which may limit water rights holders ability to benefit from longer lead times.

To address the limitations of both the Stat-PCR and Stat-P&S models, a coupled, sequential forecast approach is adopted which

utilizes both the Stat-P&S and Stat-PCR models in the following manner:



Step 1. The Stat-P&S model issues a May 1st categorical forecast of ONDJ streamflow when the Niño 3.4 conditions are met. Otherwise no forecast is issued.

Step 2a. If the Stat-P&S model issued a May 1st forecast, the Stat-PCR model re-evaluates this prediction on September 1st forecast, updating the forecast as necessary.

Step 2b. If the Stat-P&S model *did not* issue a May 1st forecast, the Stat-PCR model produces the first forecast on September 1st.

For performance evaluation, a categorical hit by Stat-P&S model becomes a miss if Stat-PCR model predicts a different (and wrong) category. The Stat-PCR model may also correct a categorical miss by the Stat-P&S model.

The May 1st Stat-P&S and September 1st Stat-PCR coupled forecast model revels a large degree of categorical forecast consistency (change between Table 4 and Table 3.) The Stat-PCR model only predicts a different category than the Stat-P&S

model in two of the 39 years evaluated, and for these two cases, it changes extreme misses (least desirable outcome) to hits. One such change was for the year 1995, one of the three lowest years of ONDJ streamflow not correctly categorized by the Stat-P&S model (initially predicted Above-normal while Below-normal streamflow observed.) Thus, the coupling of these two Stat models appears to perform superiorly as compared to models individually by skillfully increasing the prediction lead time and allowing for prediction updating, as necessary.

**3.4 Allocation Prediction Model**

A streamflow prediction-reservoir water balance model system is used to evaluate the performance of water right allocations, as compared with using streamflow observations and streamflow climatology, for a September 1st issuance. Utilizing streamflow observations is synonymous with a perfect forecast. The system is tested in hindcast mode using streamflow median and 80th percentile streamflow prediction scenarios of ONDJ streamflow separately. Both the median and 80th

percentile approaches outperform climatology, achieving Hit Scores of 53%, as compared with only a 30% Hit Score using climatology (Table 5.) Additionally, the climatological median fails to predict any years with Extreme reductions (< 0.25 liters per second); the climatology-based approach over-allocates in 55% of years, as opposed to only 27% of years when applying the 80th percentile forecast approach. This is noteworthy from a management perspective, as over-allocation is often considered more problematic than under-allocation from a long-term, drought-focused perspective. The distributions of

forecast-based allocations also more closely match observations than climatology, with the median and the 80th percentile forecast scenarios exceeding observation-based allocations by only 0.06 and 0.04 liters per second, respectively, on average (Fig. 8(a.) Over-allocation using climatological streamflow is again evident, as the interquartile range (IQR) of climatological allocations does not align with observations. While the IQR of the forecast-based scenario is larger than the observation-based



scenario, it does not systematically over-allocate (Fig. 8(a.) This can also be illustrated by calculating the ratio of each approach (climatology and forecasts) to observed allocations (Fig. 8(b.) In this case, a perfect score would be a consistent value of one, as a climatological or forecast allocation would match each observation-based allocation. The forecast-based allocation ratios produce smaller IQRs and lower median values than climatology-based allocations, implying that the forecasts
5 are better aligned with observations and slightly more conservative.

**Table 5: Categorical water right allocation results: observed vs forecast**

| Median Forecast Hit Score 53% Extreme Miss 5% | | Forecast – Sep | | |
| --- | --- | --- | --- | --- |
| | | Extreme | Severe | Moderate |
| | | < 0.25 L/s | ≤0.5 L/s | ≥0.5 L/s |
| Observed | Extreme | 10 | 11 | 1 |
| | Severe | 5 | 8 | 9 |
| | Moderate | 2 | 3 | 17 |

| 80th Percentile Forecast Hit Score 53% Extreme Miss Score 3% | | Forecast – Sep | | |
| --- | --- | --- | --- | --- |
| | | Extreme | Severe | Moderate |
| | | < 0.25 L/s | ≤0.5 L/s | ≥0.5 L/s |
| Observed | Extreme | 11 | 11 | 0 |
| | Severe | 6 | 9 | 7 |
| | Moderate | 2 | 5 | 15 |

| Climatology Hit Score 30% Extreme Miss Score 2% | | Forecast – Sep | | |
| --- | --- | --- | --- | --- |
| | | Extreme | Severe | Moderate |
| | | < 0.25 L/s | ≤0.5 L/s | ≥0.5 L/s |
| Observed | Extreme | 0 | 21 | 1 |
| | Severe | 0 | 9 | 13 |
| | Moderate | 0 | 11 | 11 |





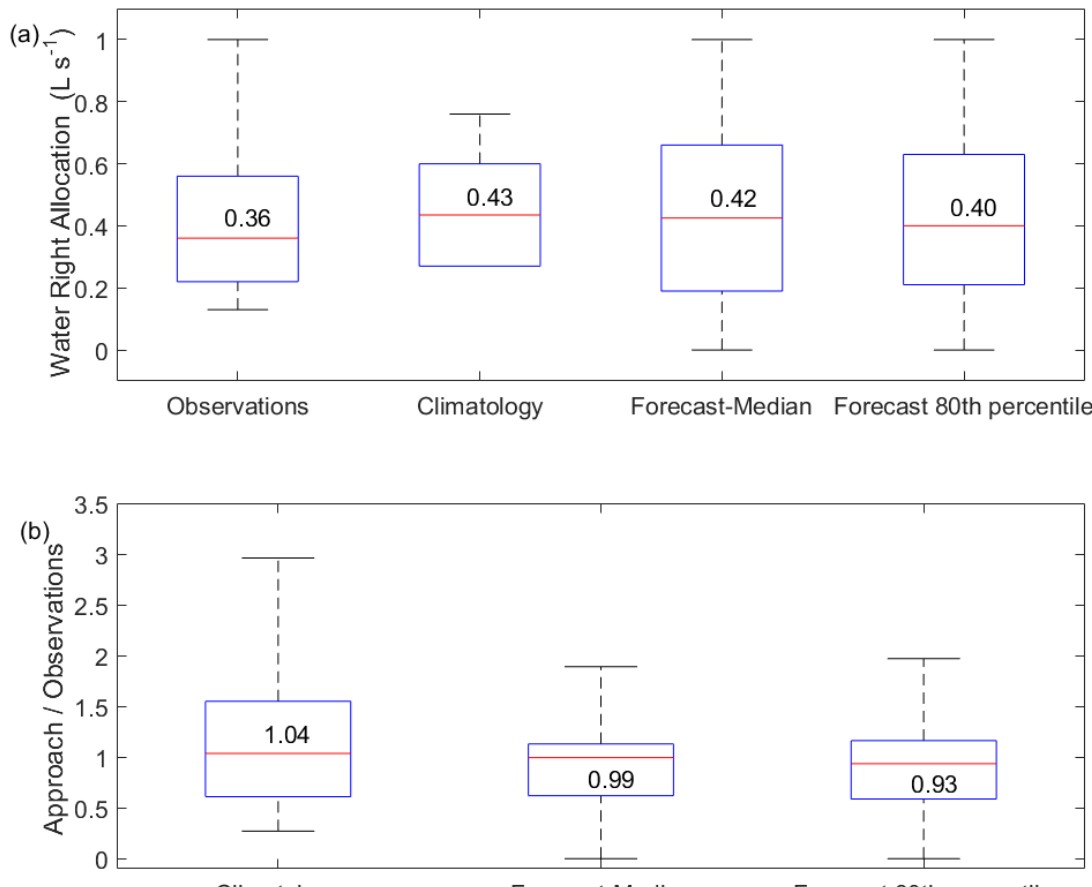

**Figure 8: Reservoir model-derived forecast allocations: (a) absolute allocation values, (b) ratio of forecast allocations to observed allocations**

5    The probabilistic modeling approach also allows for an understanding of categorical forecast certainty and strength, that is, the degree to which the model suggest a category (Fig. 9.) In this case, the forecast-based allocations more often indicate a stronger forecast tendency (higher probability) toward one category, whereas the climatology-based allocations often indicate a weaker tendency to shift. While this is not always the case, from a reservoir management perspective, climatology-based allocations provide less actionable information, as the strength of the predicted categories are often not too dissimilar, even in years where

10    correct predictions are made. In contrast, for the 28 years where forecast-based allocations of a category exceed 80% (a strong prediction), the Hit Score is 79%, a high success rate, and further, no extreme misses occur (Moderate category predicted, Extreme category observed), avoiding over-allocation in dry years.





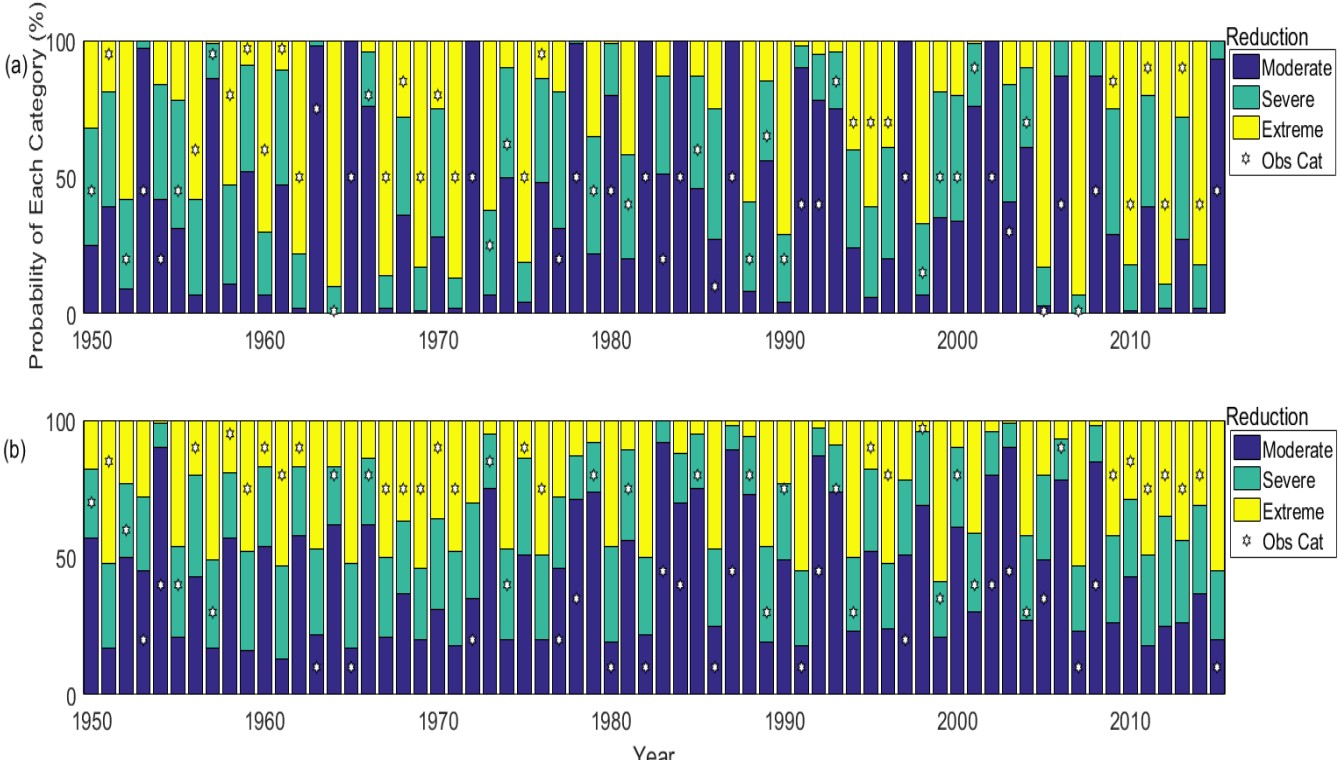

**Figure 9: Probabilistic water right allocation forecast using (a) September 1st PCR-Stat model 80th percentile, (b) long-term averages (climatology)**

5    The effect of over- and under-allocation by both forecast- and climatology-based approaches on end of year reservoir storage is also evaluated. Large deviations from the 100 million cubic meter target volume (February 1st) are viewed as problematic to the JVRE and water rights holders (Fig. 10.) The prior analysis demonstrates the propensity for the climatology-based approach to consistently over-allocate, resulting in reservoir volumes consistently below the target. The forecast-based scenarios have a smaller IQR with median values approaching the target value. The climatology-based approach also allocates

10    the full reservoir volume in 33% of years (leaving the reservoir empty), which happens in only 11% of years under the forecast-based scenarios, due to prediction error (Fig. 10.)



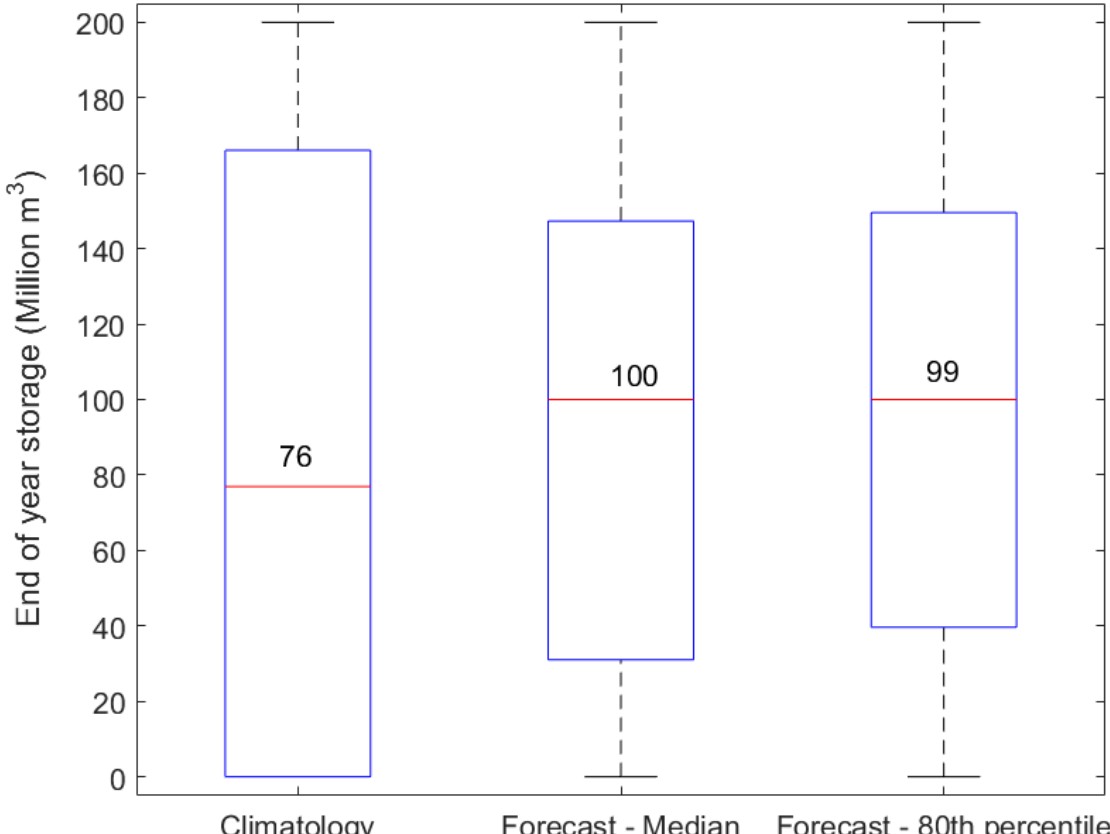

**Figure 10: End of year reservoir storage under three allocation approaches; 100 M m³ is the target**

## 4 Summary and Discussion.

The dynamic nature of ocean, atmosphere and terrestrial interactions, which contribute to moisture transport in the Elqui Valley, are undoubtedly complex and challenge hydrologic prediction models at increasing leads. The mixed success of streamflow forecasts currently in use for the Elqui reflect this. In this work, correlation and composite mapping suggest moisture transport to the Elqui Valley is dependent on the phase, strength and timing of many variables (Fig. 4.) While austral winter precipitation, SST, and soil moisture correlations with ONDJ streamflow at varied leads are encouraging (Fig. 5), the Stat-PCR approach, which makes use of these predictors, is skillful only at a September 1st lead, as indicated by RPSS scores and other forecast validation metrics (Table 2.) The Stat-Dyn approach, using precipitation and SSTs, results in inferior outcomes compared with the Stat-PCR model. The Stat-P&S model, however, provides skillful predictions of ONDJ streamflow at a May 1st lead, albeit categorically and is triggered in only 60% of the period 1950 – 2015.



The coupling of the Stat-P&S and Stat-PCR models to produce initial (May 1st) and updated (September 1st) forecasts may be valuable to both reservoir managers and water rights holders. From a reservoir management perspective, properly setting the

per right water allocation (September 1st) is critically important to satisfy rights holders and maintain adequate reservoir storage for the uncertain future. The Stat-PCR component of the coupled model provides skill superior to climatology, and is perhaps sufficient to inform these decisions. Reservoir managers, however, are also expected to provide a non-binding May 1st forecast, upon which rights holders, specifically farmers who have crop choice flexibility and/or water right leasing potential, may choose to utilize in preparation for the ONDJ growing season (e.g. determine whether to supplement through the water market.)

The Stat-P&S categorical forecast with a May 1st lead can inform these longer planning actions. The strong categorical consistency between the May 1st Stat-P&S and September 1st Stat-PCR forecasts may also serve to reinforce confidence in the forecast outcomes; the two models only differ in prediction categories twice in 66 years.

The Stat-PCR streamflow prediction-reservoir water balance model system produces values closely matched with

observations, and each forecast (median, 80th percentile) outperforms climatology. Use of the 80th percentile Stat-PCR forecast is intended to represent risk aversion; however, the probabilistic framework allows assessment for any risk preference. Ensemble predictions illustrate the general propensity of a climatology-based allocation to provide limited actionable information in contrast to forecast-based allocations, which exhibit enhanced skill when the model issues strong predictions (>80% categorical likelihood.) However, in years when the Stat-PCR forecast-based allocation model issues a weak prediction

(no dominant tendency toward any specific category) other allocation decision frameworks may be worth investigating (e.g. allocation based on existing storage only as a hedge against inflow uncertainty.)

Selection of categorical thresholds is based on equal distribution of observations, and does not necessarily represent the preferences of reservoir managers, however these thresholds can be updated. For example, if only two categories are used,

allocations above and below 0.75 L/s, the Hit Score rises to 92%. Managers are thus free to select categories which suit their needs or reflect true differences in the utility of allocations to water rights holders.

The focus of this research is to develop an understanding of the mechanisms contributing to austral summer streamflow in the Elqui Valley, investigate model skill at varied forecast leads, and produce forecast-based water-right allocations to inform

water resources management decision-making. While the approaches in this research are mostly a demonstration of concept, the model framework is consistent with the current operations of Puclaro Reservoir, but is not optimized for or hedge against expected future conditions. While the model may be informative over the long-term, resulting in allocation and storage values better matched to observations than climatology-based allocations, it performs poorly in certain years, most notably during the 2009 – 2015 hydrologic and meteorological drought (Fig. 9(a.) While poor model performance during this period is

undoubtedly due in part to the limited reservoir operating rules, the Stat-PCR approach tends to under predict extremes, especially when they occur consecutively. Further forecast model development will focus on improving predictive skill of extreme events, particularly dry periods, making use of non-parametric methods and additional multi-model approaches, and dynamic rule structures and simulation techniques. Even so, adoption of the approaches presented here by water managers

and rights holders bodes well for improved economic efficiency and benefits across the Elqui Valley.

**Code Availability.**

Should future reproduction of results become necessary, any codes will be made available, by the corresponding author, upon request.

**Data Availability.**

The data used to produce this research come from open sources, including the Chilean Ministry of Public Works – Dirrecion de Aguas (DGA) and the National Oceanic and Atmospheric Administration. Through use of the International Research Institute's Data Library, all relevant data sets may be obtained.

**Appendices.**

None.

**Supplemental Link.**

To be included by Copernicus

**Team List.**

Justin Delorit
Edmundo Cristian Gonzalez Ortuya
Paul Block

**Author Contribution.**

Justin Delorit, Edmundo Cristian Gonzalez Ortuya, Paul Block each contributed to the hydroclimatological analysis, developed model code and evaluated simulations.




**Competing Interests.**

None.

**Disclaimer.**

To be added later.

**Acknowledgements.**

This work is partially funded by a scholarship provided by the Air Force Institute of Technology.

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
