# Peer review of "Evaluation of model-based seasonal streamflow and water allocation forecasts for the Elqui Valley, Chile"

_Hydrology and Earth System Sciences, 2017_

## Referee Comment (RC1) · Anonymous Referee #1 · 22 Mar 2017

**Review of the manuscript "A Framework for Advancing Streamflow and Water Allocation Forecasts in the Elqui Valley, Chile", by Delorit *et al.***

The manuscript discusses an approach to provide actionable seasonal climate information about precipitation and streamflow for the Elqui Valley, in Chile. The results are robust, and they will be of use for both the scientific and the climate service communities. I recommend to publish it in HESS after the authors have addresses several (minor) issues.

Specific Comments

P1,L8-16: The abstract should clearly include concrete results. The paper is far more interesting than what the abstract suggests. For example, a mention of the way the models are used in sequential mode, or the skill achieved, can be mentioned there.

P1L20: "institutions"

P4,L13,L16: something is wrong with the way the references are being written. E.g., it should be Aceituno, 1988.

P7, Section 2.1: Is the data quality-controlled? Maybe add a sentence or two about that, so the reader knows if the data can be trusted.

P7,L17,L21 (and other places): vector winds? This is the first time I see that name. What the authors mean by that? To use both u and v? Why do not just say "winds"?

P8L8: why to do a spatial average? I do not fully understand that sentence.

P9L1: are the authors talking about CFSR? They are talking about NCEP-NCAR reanalysis but then they cite Saha et al 2013. Is there a confusion here?

P10: define N.

P11L21: why a forecast is not issued in that case? Explain in the text.

P11Section2.2: I suggest to change the title of the subsection, as it seems to be about a proper dynamical prediction model, and it is really about using dynamical model output in a statistical model.

P11L28: I think the authors mean "North American Multi-Model Ensemble".

P12L8: authors should be a bit more explicit about when local variables have predictive strength. Conditions? Dates? Proportion of total cases? More information is needed.

P14L1-2: please check the syntax of the sentence.

P14, eqs 3 and 4: how are these equations obtained?

P14Section2.4: I suggest to remind the reader that all these results are obtained using cross-validation (a lot of studies out there do not even bother to cross-validate!)

P14L9: why Pearson coefficient?

P14L10: …as opposed to *a specific quantity*….

P16L2: references in capital

P16L22-24: confusing sentence… too many commas?

P17L11: maybe change "affirming" → "confirming"?

P19L20: approach or model? Which one?

P19L20-21: I do not understand the sentence. When the other 40% occur?

P20Step2a and Step2b: what is the real difference here?

P25L25: 92% is extremely high. Can you please confirm there is not a typo there?

---

## Author Comment (AC1) · 3 Apr 2017

**Reply to Anonymous Referee 1**

We thank Referee 1 for reviewing our manuscript and providing constructive, actionable feedback. Below we provide our responses to each point raised.

**Comment:** P1, L8-16: The abstract should clearly include concrete results. The paper is far more interesting than what the abstract suggests. For example, a mention of the way the models are used in sequential mode, or the skill achieved, can be mentioned there.

**Reply:** This is an excellent comment, and we have added the following text to the abstract of the working manuscript (P1, L16-27):

"Skillful results (forecast outperforming climatology) are produced for short lead-times (September 1st; RPSS = 0.31), where categorical hit skill is 61%, with Above (wet) and Below-Normal (dry) flow years achieving 82% and 64% categorical hit skill, respectively. At longer lead-times, climatological skill exceeds forecast skill, largely due to less observations of precipitation in the statistical model (August 1st RPSS = 0.02 and July 1st RPSS = -0.39). Coupling the September 1st statistical forecast model with a Niño 3.4 region sea surface temperature phase and strength statistical model allows for equally skillful categorical streamflow forecasts to be produced for a May 1st lead, triggered for 60% of the years in the period 1950-2015. The reservoir allocation model is skillful at the September 1st lead (categorical hit skill score = 53%), and using a probabilistic modeling approach, forecast-based allocations are categorically skillful (79%) when the model predicts the observed allocation category with at least 80% confidence ($\geq$ 80% of annual forecast values fall within a single category). The frameworks applied here advance the understanding of the mechanisms and timing responsible for moisture transport to the Elqui Valley, and provide a unique application of streamflow forecasting in the prediction of per-water right allocations. Both have the potential to inform water right holder decisions."

**Comment:** P7, Section 2.1: Is the data quality-controlled? Maybe add a sentence or two about that, so the reader knows if the data can be trusted.

**Reply:** Observations of streamflow and snow water equivalent are obtained from the Direccion General de Aguas (DGA), a department of the Ministry of Public Works of the Chilean Government. Collection, validation and quality control of hydrologic measurements are part of DGA's core functions; thus, we treat the data as fully vetted and having met DGA's quality control standards. The referee's comment is valid, and warrants an addition to the manuscript. We have added the following (P7 L15-18):

"One of DGA's primary functions as the regulator of surface water resources for the Chilean Government is to collect, validate, and perform quality control of hydrologic measurements. Open source data obtained through DGA is considered as having met DGA quality standards."

**Comment:** P7, L17, L21 (and other places): vector winds? This is the first time I see that name. What the authors mean by that? To use both u and v? Why do not just say "winds"?

**Results:** We are concerned with both the magnitude (colors) and direction (arrows are the resultant of u and v) of vector winds at 500mb (Fig. 4. (a) and (b), *excerpt below*). Both are critical in terms of determining the efficiency of moisture transport to the Elqui Valley. If we ignore the direction component, "winds" or "wind speed" should be used. Additionally, "vector winds" is the name commonly utilized by NOAA and other climate agencies.

[Figure]

**Comment:** P8L8: why to do a spatial average? I do not fully understand that sentence.

**Reply:** Gridded potential predictors are identified through spatial composite and/or correlation mapping (e.g. sea surface temperatures (SSTs). To extract the signal(s) from within the gridded data set and avoid noise present at the grid scale, principal component analysis (PCA) is commonly applied to the gridded data. Correlating the first principal component (PC), which is the strongest signal, with the spatially averaged data identifies whether the signal is spatially homogenous. Alternatively, if the first PC does not correlate well with the spatial average, the heterogeneity of the dataset is important, and thus using the spatial average may not be the best approach. For example, the spatial average of SSTs (Fig. 4 (c)), which is identified as a potentially significant as predictor of streamflow for the Elqui River (roughly consistent with the Niño 3.4 region), correlates highly (>0.9) with the first PC of the gridded SST data. Having identified SSTs as spatially homogenous, and consistent with the Niño 3.4 region, we correlate and ultimately select the Niño 3.4 Index as a potential predictor of streamflow as it is well-known, well understood, and well-studied. We do this as opposed to selecting a marginally different (e.g. sub-region of the Niño 3.4 region), but much less understood and perhaps less defensible area. Furthermore, using the spatial average rather than

selecting a sub-regional area may be a more conservative approach as it does not guarantee that the strongest possible relationship is identified. In addition, using an index avoids grid cell selection bias (cherry-picking), which could result in an insufficient number of grid cells to be statistically significant, or produce vastly different regions of high correlation (spurious correlations.) The following papers support the claim of teleconnections between precipitation and SST in north-central Chile (Aceituno 1988; Falvey and Garreaud 2007; Garreaud et al. 2009; Montecinos and Aceituno 2003).

[Figure]

**Comment:** P9L1: are the authors talking about CFSR? They are talking about NCEP-NCAR reanalysis but then they cite Saha et al 2013. Is there a confusion here?

**Reply:** We appreciate the reviewer catching this improper citation. Clearly, the citation should be (Kalnay et al. 1996) as opposed to Saha et al 2013. We also add (Huang, van den Dool, and Georgarakos 1996) to specifically reference CPC's soil moisture data. The working manuscript has been updated with the appropriate citations.

**Comment:** P11L21: why a forecast is not issued in that case? Explain in the text.

**Reply:** The statistical phase and strength model (Stat-P&S) at the May lead does not provide a categorical streamflow forecast for Niño 3.4 Index = (+0.5°C, +0.75°C) or (-0.5°C, -0.75°C) as the range is considered transitional (not weak or moderate as identified by NOAA). Both the magnitude and persistence of SST observations in this range do not allow for production of skillful forecasts. The May 1st forecast lead uses January-April Niño 3.4 Index values to categorically forecast October-January streamflow, which requires prediction through the Spring Barrier (Duan and Wei 2013). Typically, SSTs

within the transitional range are not stable and actively moving to either a neutral or strengthened phase. Until these changes occur, at some date beyond May 1st (typically beyond the Spring Barrier), a categorical or deterministic forecast is typically not skillful. Deferring to the September 1st statistical principal component regression model (Stat-PCR) is warranted when SSTs are in the transitional range.

The question is valid, and our original manuscript does not address the reasons for which the transitional range within the Niño 3.4 Index is not used by the Phase and Strength model at the May lead. To provide clarity in our approach we have added the following to the working manuscript (P11 L21-24):

"For these ranges, neither the magnitude (not weak or moderate as defined by NOAA) nor persistence of SST observations allow for production of skillful categorical streamflow forecasts. For years where SSTs fall within these ranges at forecast leads prior to the Spring Barrier, strength and phase are subject to rapid transition, and categorical forecasts are typically not skillful."

**Comment:** P11Section2.2: I suggest to change the title of the subsection, as it seems to be about a proper dynamical prediction model, and it is really about using dynamical model output in a statistical model.

**Reply:** Regarding dynamical model prediction, we initially considered raw dynamical climate model outputs of precipitation and SSTs to predict streamflow (since clearly streamflow is not an output of dynamical climate models), but the results were poor. We thus proceeded with statistical post-processing as a means of correcting dynamical model outputs (Gheti 2008). The reviewer's point is valid as the sub-section title may be interpreted as dynamically modeled streamflow, including a physically-based hydrology model. To avoid confusion and as a means of accurately describing the forecast approach we have changed the title of 2.2 to "Hybrid dynamical-statistical streamflow prediction model" to capture the fact that predictors come from the dynamical model, but the prediction model formulation is still statistical in nature.

**Comment:** P12L8: authors should be a bit more explicit about when local variables have predictive strength. Conditions? Dates? Proportion of total cases? More information is needed.

**Reply:** Local variables and their predictive strength are discussed in 2.1, and shown in Figure 5 (a) and (c) and in Table 1. The same variables are used, when appropriate, for the statistical model using corrected (quantile mapping) GCM outputs for precipitation and SSTs (Stat-Dyn), with forecasts issued January 1st , May 1st and June 1st. For these leads, local variables are not useful, and therefore only GCM predictions of precipitation and SSTs are used (Table 2. *Excerpt below*). This is not a surprising result considering local variables are skillful in the prediction of October-January streamflow only during months of peak precipitation (May-August) as shown in the manuscript in Figure 5 (a) and (c).

Still, we recognize readers may benefit from additional explanation and have added the following for clarity of local variable inclusion in the working manuscript (P12 L10-13):

"The Stat-Dyn model is meant to provide streamflow forecasts at extended leads, beyond what is possible with global and local observed data used to inform the Stat-PCR model. Local variables (e.g. precipitation, snow water equivalent and soil moisture) hold the most predictive strength during the season of peak precipitation (May-August) and thus are only considered for the Stat-Dyn model for leads at prior to June 1st (Fig. 5 (a.) and (c.)."

| Forecast | | Retained Predictors | | |
|---|---|---|---|---|
| Statistical Approach (Stat-PCR) | Sep 1st | Aug SM | JA Prcp | Aug 3.4 |
| | Aug 1st | Jul SM | JJ Prcp | Jul 3.4 |
| | Jul 1st | Jun SM | MJ Prcp | Jun 3.4 |
| Dynamical Approach (Stat-Dyn) | Jun 1st | JJA 1.2 | JJA Prcp | - |
| | May 1st | JJA 3.4 | JJA Prcp | - |
| | Jan 1st | JJA 3.4 | | - |

**Comment:** P14L1-2: please check the syntax of the sentence.

**Reply:** We agree the structure of the sentence can be improved to better illustrate the point. We have changed the sentence in the working manuscript to:

"Allocation, as issued annually by JVRE, and storage outcomes are hindcast in a cross-validated mode for the period of record (1950 – 2015) by coupling the streamflow prediction models to a simple reservoir balance model."

**Comment:** P14Section2.4: I suggest to remind the reader that all these results are obtained using cross-validation (a lot of studies out there do not even bother to cross-validate!)

**Reply:** We thank the referee for the comment, and have included language which reminds the reader the forecast outputs are cross-validated.

**Comment:** P15L9: why Pearson coefficient?

**Reply:** Pearson's correlation coefficient is commonly used to assess both the general parametric association between forecast and observed values, and phase error. While it doesn't account for forecast bias and is sensitive to outliers, it is selected because it is well known and well understood. In addition, we utilize RPSS and categorical skill score metrics which describe additional performance and features of the forecasts.

**Comment:** P19L20: approach or model? Which one?

**Reply:** We appreciate the referee noticing and highlighting this error. For consistency, we use "model".

**Comment:** P19L20-21: I do not understand the sentence. When the other 40% occur?

**Reply:** 40% refers to a fraction of the number of years in the record (1950-2015) not predicted by the Stat-P&S model at the May lead using January-April Niño 3.4 Index because the index values fall within the transitional ranges (+0.5°C, +0.75°C) or (-0.5°C, -0.75°C). The transitional ranges do not provide skillful categorical forecasts for the May 1st lead. For this reason we do not forecast these years until the Stat-PCR model is skillful for the September lead. Our coupled statistical prediction model defers prediction for these years to September.

**Comment:** P20Step2a and Step2b: what is the real difference here?

**Reply:** The difference between Step 2a and 2b relates to whether the Stat-P&S model issues a May forecast. If January-April Niño 3.4 region SSTs meet the Stat-P&S criteria, a May 1st categorical forecast is issued (Step 2a). Otherwise, the Stat-PCR model is used to produce a September 1st forecast (Step 2b). The novelty of coupling the Stat-P&S and Stat-PCR models is the Stat-P&S model provides an initial, categorical indication (May 1st lead) of October-January streamflow for 60% of years between 1950-2015. From the perspective of a water rights holder, a skillful categorical forecast at a May 1st lead may provide relevant information to inform October-January decision making (e.g. cropping decisions by water right holding farmers). The initial forecast is reinforced by the Stat-PCR model, which provides a skillful deterministic forecast, but only for a September 1st lead. The key is that the agreement between the May 1st categorical forecast produced by the Stat-P&S model (when issued) and the September 1st deterministic Stat-PCR model is very high. In applied terms, a water right holder seeking to make forecast informed decisions has information which outperforms climatology at leads up to 4 months prior to the issuance of the actual allocation value.

**Comment:** P25L25: 92% is extremely high. Can you please confirm there is not a typo there?

**Reply:** We agree, 92% is a very high 'hit score'. However, this is a calculation of how often the forecast category aligns with the observed category, and is not a correlation. The hit score referenced here is for two categories, divided at 0.75 L s$^{-1}$ as opposed to three categories as discussed at length in the paper. This case provides less overall information (probability of the allocation above or below the threshold) and thus we are not overly surprised that the score increases dramatically. In fact, we select this case specifically to compare with the three-category allocation model, to illustrate how categorical skill is a product of the bounds selected by the stakeholder.

**Comment:** P14, eqs 3 and 4: how are these equations obtained?

**Reply:** The purpose of the reservoir allocation model is to compare allocation and storage outcomes from the forecast and climatology informed reservoir operations against observations (which constitute a perfect forecast); this provides a means of evaluating the streamflow forecast in an applied context. Thus, equations 3 and 4 are simply a modified version of the reservoir balance. To include the annual, end-of-year storage target used by reservoir operators in the Elqui (100 million cubic meters), we adjust allocation for period $i+1$ by the storage deficit or surplus at the end-of-year $i$. For example, if the forecast informed allocation, $A_{i_{ONDJ}}$, results in an end-of-year storage $S_{i_{Feb}} \leq 100M\ m^3$, $A_{i+1_{ONDJ}}$ is penalized by the difference $100 - S_{i_{Feb}}$. In contrast, if $S_{i_{Feb}} \geq 100M\ m^3$, $A_{i+1_{ONDJ}}$ is boosted by the absolute value of the difference of $100 - S_{i_{Feb}}$. It is important to note that the equations are applied uniformly to the forecast, climatology, and observations, so a fair assessment of performance can be achieved.

**Comment:** P16L22-24: confusing sentence… too many commas?

**Reply:** We thank the referee for this comment, and agree that the sentence is confusing. We have replaced it the working manuscript with the following:

"As forecast lead increases, both Hit Score and RPSS decrease, while Extreme Miss Score increases. These results occur because less information regarding the MJJA rainy season is available, which is consistent with decreased correlations between ONDJ streamflow and predictors (Fig. 5.)"

**Minor Comments: (all accepted and corrected in the manuscript)**

P1L20: "institutions"

P4, L13, L16: something is wrong with the way the references are being written. E.g., it should be Aceituno, 1988.

P10: define N.

P15L10: …as opposed to *a specific quantity*….

P11L28: I think the authors mean "North American Multi-Model Ensemble".

P16L2: references in capital

P17L11: maybe change "affirming" to "confirming"?

**References:**
Aceituno, Patricio
 1988  On the Functioning of the Southern Oscillation in the South American Sector. Part I: Surface Climate. Monthly Weather Review 116(3): 505–524.

Duan, Wansuo, and Chao Wei
 2013  The "spring Predictability Barrier" for ENSO Predictions and Its Possible Mechanism: Results from a Fully Coupled Model. Internation Journal of Climatology 33: 1280–1292.

Falvey, Mark, and René Garreaud
 2007  Wintertime Precipitation Episodes in Central Chile: Associated Meteorological Conditions and Orographic Influences. Journal of Hydrometeorology 8(2): 171–193.

Garreaud, René D., Mathias Vuille, Rosa Compagnucci, and José Marengo
 2009  Present-Day South American Climate. Palaeogeography, Palaeoclimatology, Palaeoecology 281(3–4). Long-Term Multi-Proxy Climate Reconstructions and Dynamics in South America (LOTRED-SA): State of the Art and Perspectives: 180–195.

Gheti, Rares
 2008  Statistical Post-Processing of Dynamical Surface Air Temperature Seasonal Predictions Using the Leading Ocean-Forced Spatial Patterns. http://digitool.library.mcgill.ca/R/?func=dbin-jump-full&object_id=18670&local_base=GEN01-MCG02, accessed March 30, 2017.

Huang, Jin, Huug M. van den Dool, and Konstantine P. Georgarakos
 1996  Analysis of Model-Calculated Soil Moisture over the United States (1931–1993) and Applications to Long-Range Temperature Forecasts. Journal of Climate 9(6): 1350–1362.

Kalnay, E., M. Kanamitsu, R. Kistler, et al.
 1996  The NCEP/NCAR 40-Year Reanalysis Project. Bulletin of the American Meteorological Society 77(3): 437–471.

Montecinos, Aldo, and Patricio Aceituno
 2003  Seasonality of the ENSO-Related Rainfall Variability in Central Chile and Associated Circulation Anomalies. Journal of Climate 16(2): 281–296.

---

## Referee Comment (RC2) · Anonymous Referee #1 · 10 Apr 2017

I have reviewed the changes made to the manuscript and I'm almost completely satisfied with the improved version. I recommend to accept it for publication in HESS after very minor changes have been performed.

I just want to suggest the authors to actually include at least part of their explanation in the manuscript, so other readers can take advantage of what they wrote in this review (even if the review is public). I think this is only necessary in these two cases in the original version:

1) P8L8 – on the spatial average.

2) P19L20-21 –on the 40%.

[Figure]

Finally, I still think it is better to call them "wind vectors" rather than "vector winds", but I am not asking the authors to make any change about that.

I do not need to review the manuscript again.

---

## Author Comment (AC2) · 12 Apr 2017

**Reply #2 to Anonymous Referee 1**

We thank Referee 1 for reviewing our responses and once again providing valuable feedback. Below we provide our responses to each point raised.

**Comment:** P8L8 – on the spatial average. (Add text to manuscript)

**Reply:** To further clarify the purpose for and process by which spatial data is averaged we have added the following to the working manuscript (P8, L8 – P9, L5):

"The first principal component (PC) from the gridded variable region, representing the dominant signal in the gridded field, correlated with the spatial average of the gridded variable region can identify if the signal is spatially homogenous (representative) across the region. If the first PC does not correlate well with the spatial average, the heterogeneity of the dataset is likely important, and adopting the spatial average as a predictor may be ineffective. For example, the spatial average of SSTs (Fig. 4 (c.)), a potentially significant predictor of streamflow for the Elqui River, correlates highly (>0.9) with the first PC of the gridded SST data. This region of SSTs is closely aligned with the quintessential ENSO pattern in the equatorial Pacific Ocean, and is evident when correlating the entire ONDJ streamflow record with SST anomalies in the preceding MJJA, which suggests ENSO, in general, plays some role in explaining streamflow variability within the Elqui Valley (Fig. 4(c.)) Having identified SSTs as spatially homogenous, and consistent with the Niño 3.4 region, we select the Niño 3.4 Index as a potential predictor of streamflow, in lieu of the SST region initially identified (Fig. 4c), as it is well-known, well understood, and well-studied."

**Comment:** P19L20-21 –on the 40%. (Add text to manuscript)

**Reply:** To further clarify the Stat-P&S model criteria we have added the following to the working manuscript (P20, L21-25):

"These ranges are transitional and do not provide skillful categorical forecasts for the May 1st lead. For this reason, the coupled statistical prediction model defers prediction for these years to September 1st, when the Stat-PCR model is skillful." The Stat-PCR approach provides deterministic forecasts of ONDJ streamflow, it is only skillful at a September 1st forecast lead, which may limit water rights holders ability to benefit from longer lead times."

**Comment:** It is better to call them "wind vectors" rather than "vector winds", but not asking the authors to make any change about that.

**Reply:** To avoid any confusion, we provide reference to both vector winds and wind vectors in the manuscript. The term "vector winds" is common amongst the climate community, thus we have opted to retain it as well. (P7, L20):

"…vector (also referred to as wind vectors) and meridional winds…."

---

## Referee Comment (RC3) · Anonymous Referee #1 · 13 Apr 2017

I'm now satisfied with the authors' revision of the manuscript and I recommend it for publication in HESS.

---

## Referee Comment (RC4) · Anonymous Referee #2 · 26 Apr 2017

This manuscript describes the implementation and evaluation of a streamflow forecasting tool for an arid basin with a high water demand in northern Chile. Three types of forecasting models are used, two of which are referred to as "statistical", and one as "dynamical". These models are then coupled to a simple water allocation model, and evaluated by running them with historical input data in hindcast mode.

The first model uses principal component regression to fit various large-scale meteorological variables against streamflow. The second model is simpler and only uses an ENSO index. A third model uses precipitation data from the NCEP climate forecast system (though it is unclear what lead times are used) and relates these to streamflow

using a quantile mapping approach.

These models are then run with lead times between 1 and 5 months, and linked to water allocation using a water allocation model.

The study finds "mixed success" in the models' capacity to predict streamflow and related water allocation levels.

The manuscript is generally well-presented and well-written and most of the figures are of good quality. However, with regard to the content, I identify many issues of varying severity. But all together I think that they warrant a very serious revision of the study.

First, the scientific contribution is currently unclear. To be blunt, the study currently reads as a consultancy report, with an extensive description of the study case, and a very elaborate description of the model implementation, but very limited scientific contextualization and discussion. A scientific study should be different. Rather than solving a specific issue for a specific location (as I think this study currently does), a case study should be used to gain broader insights in hydrological processes, modelling concepts, and/or improvements of existing modelling tools and methods. Here, the current manuscript falls short in my opinion. This starts with the title. What is meant with a "framework" in this context? Essentially, the approach couples a streamflow forecasting model to a water allocation model. That is a sensible, but not particularly novel approach, and can hardly be considered a specific modelling framework. Similarly, the word "advancing" is probably redundant. I think that a more appropriate title would be "Evaluating model-based seasonal streamflow forecasting for the Elqui Valley, Chile".

Next, the study does not have a clear scientific question or (ideally) hypothesis. As highlighted on p.25 line 28 - 30, it intends to "develop an understanding of the mechanisms contributing to austral summer streamflow in the Elqui Valley, investigate model skill at varied forecast leads, and produce forecast-based water-right allocations". That is of course very broad and vague, and as a result, the study does not really make an impact on any of them. As for the first objective, I do not think that I gained much

insight in climatic teleconnections with streamflow beyond what is quite well known. As for investigating model skill, I am not sure what conclusions can be drawn that have relevance beyond the particular case study. And I am surely happy to believe that producing forecast-based water allocations is of great local interest, but again the scientific value is unclear.

Furthermore, many aspects of the modelling approach remain unclear. One reason for this is the description of the data and models is intermingled, making it very hard to follow. I suggest splitting this section up, describing first the data that are used and their characteristics (e.g., coordinates, temporal and spatial resolution, source). Then, a next section can describe the modelling approach and make a clear argumentation as to why those specific models were chosen. This is important, because the selection and design of models is odd. Why, for instance, was a model built that only uses the ENSO index? Clearly this will have limited predictive capacity. If the purpose is to evaluate the predictive capacity of the ENSO index (though I am not sure why one would want to do this) then it is probably better to set up a specific statistical model such as a Generalized Linear Model, which allows for a more rigorous evaluation of statistical significance and predictive power of different predictors.

Also for a problem like this, the most obvious approach to streamflow forecasting would seem to be to route precipitation forecasts or observations (depending on the lead time) through a hydrological model. Seasonal forecasts are globally available at increasing resolution (well above the 250 - 600km mentioned on p.11 l.24 - the reference of Giorgi (1990) is probably out-of date!). Additionally, because of the large time lag between precipitation and streamflow it may well be possible to use observed precipitation, especially for the shorter lead times (August, September predictions). In fact, given that streamflow is so strongly snowmelt-driven in this basin, I wonder whether observations of snow cover and snow depth during the austral winter might be variables with a very strong predictive power. All this to say that the scientific value of trying to make predictions based on large scale climatic predictors is questionable, and at least needs to be

justified much better.

Next, I also had a hard time understanding the context of the water allocation model. Why is the target reservoir volume in February 50%? This would seem very much to me. Also, the fact that any shortfall of this 50% needs to be carried over to the next year (eq. 3) suggests that the reservoir is not replenished during winter. Is this realistic? This would seem to depend strongly on winter precipitation.

As for the "summary and discussion" section, this is very thin and little informative. I suggest elaborating the "discussion" and adding a separate "conclusions" section. This is not only more usual for HESS, but I also think that the lack of conclusions may be somewhat indicative of some of the main problems identified above. If anything, adding a "conclusions" section would be a very useful exercise to think about what particular scientific conclusions can be drawn from the study.

Lastly, I think that the manuscript can be shortened. It contains too much undergraduate text book material that can be removed, e.g., on principal component analysis, Global Circulation Models, performance metrics and similar tools.

---

## Author Comment (AC3) · 12 May 2017

Reply to Anonymous Referee 2

We thank Referee 2 for carefully reviewing our manuscript and providing thoughtful feedback. Below we provide our responses to each point raised. We cite several instances where changes are made to the working manuscript, which will be submitted to HESS (if accepted to move forward) consistent with the review timeline.

**Comment:** The manuscript is limited to the study case, and as such it is not likely to help readers "gain broader insights in hydrological processes, modelling concepts, and/or improvements of existing modelling tools and methods."

**Reply:** We agree that the streamflow forecast component represents a location-specific contribution, however we believe this forecast coupled with the human managed allocation system is collectively novel and broadly relevant. While we address the unique set of circumstances posed by the Elqui Valley, Chile, the implications of the framework apply to basins where water rights represent a mechanism to promote equity and efficiency in the use of limited water resources.

The comment is valid, and accordingly we have made this point clearer in the discussion and conclusions. We address this in the working manuscript by describing how water rights driven basins might increase allocation efficiency by implementing forecasts as opposed to climatology based information as part of their decision framework.

Additional novelty lies in the coupling of the Stat-P&S and Stat-PCR forecast models, which provides a May 1st categorical prediction followed by a deterministic prediction on September 1st. The high level of agreement between the models suggests that while determinism is lost extending the lead from September 1st to May 1st, accurate categorical predictions are possible. We are not aware of work which has produced similar forecast skill at a May 1st lead in North Central Chile. This result also holds the potential for broader applications. Coupled forecasts need not be strictly deterministic, and using early categorical forecasts to provide an indication of expected conditions, and reinforcing the prediction with a revised deterministic forecast as more observations of local variables (e.g. precipitation) become available may be useful for a water rights holder.

Our discussion and conclusions section does address this point, but is lacking in terms of linking to a broader application. We address this in the working manuscript by bolstering the existing discussion.

**Comment:** The manuscript contains no clear hypothesis against which the research can be assessed.

**Reply:** This is a valid comment and is addressed in the working manuscript both in the introduction, discussion and conclusions by reframing the purpose of the research, namely to test if:

"…skillful streamflow forecasts can be coupled with reservoir allocation decision models to improve allocation efficiency as compared to climatology based decisions."

Additional discussion pertaining to the outcome of the research, as it pertains to addressing the hypothesis, is included in the discussion and conclusions.

**Comment:** Separating Section 2 into distinct 'data' and 'modelling approach' sections may make the modelling approaches more clear to the reader.

**Reply:** We are compelled to combine both the data and the modelling approaches in a common section as the data are not shared by each model. The Stat-PCR models are informed by observations of precipitation, soil moisture and sea surface temperatures, while the Stat-P&S model makes use of Niño 1.2 and 3.4 Index values, and the Stat-Dyn uses dynamical model outputs of both precipitation and sea surface temperatures. Rather than consistently refer to a data section when describing the modelling approach, we feel the logical approach is to introduce the data as it corresponds to the appropriate model and lead.

Still, we acknowledge the validity of this comment, noting that data and methods are presented separately in many peer reviewed papers. If the Referee feels strongly about separating the sections, we are happy to do so.

**Comment:** The Referee notes a lack of description of the reasons for which the modelling approaches are selected. Specifically, the Stat-P&S model, which uses Niño 1.2 and 3.4 Index values seems inappropriate, and leads are not well described.

**Reply:** We recognize the presentation of forecast leads for each model are not clearly described in the manuscript. To address this concern, we add additional language to the description of each model to clarify the leads considered, and provide supplemental text in the results section further reinforcing the leads and corresponding model skill. Additional information and detail are provided below (and added to the working manuscript.)

There are three distinct streamflow modelling approaches used in this research, aimed at balancing model skill and lead time. All are classified as statistical.

1) The Principal Component Analysis (Stat-PCR) model is meant to provide a deterministic prediction of streamflow using the most skillful and defensible predictors possible for increasing leads. Use of PCR is common in research focused on season-ahead streamflow prediction, and applying the leave-one-out cross-validated methodology adds additional credibility of the approach. Leads extend monthly from June 1st to September 1st. As described in the manuscript, observed data before June does not add to model skill.

2) The use of quantile mapping to correct dynamical model outputs of precipitation and sea surface temperatures (Stat-Dyn) is implemented in the same manner as the Stat-PCR. The main purpose of the Stat-Dyn approach is to increase the lead time of the streamflow predictions beyond what is possible with the Stat-PCR model. For example, the January 1st dynamical model outputs for May-August precipitation and sea surface temperatures are used to produce statistical streamflow predictions with a January 1st lead. Leads extend monthly from January 1st to June 1st.

3) The Phase and Strength model (Stat-P&S) makes use of the persistence of sea surface temperatures by using the Niño 1.2 and 3.4 indices, as opposed to other predictors which are shorter-lived or only become apparent at later leads (e.g. precipitation, soil moisture, pressure). The Stat-P&S approach is only use to provide a categorical prediction of streamflow, and ultimately proves more skillful at a May 1st lead than the Stat-PCR approach. May 1st is the only lead-time fir Stat-P&S.

The coupling of the Stat-P&S and Stat-PCR approach provides a skillful categorical streamflow prediction at May 1st (Stat-P&S), which is solidified by a September 1st deterministic prediction (Stat-PCR). The strength of the coupled model is the high degree of agreement between the two components. For all but two of the 39 years predicted by the Stat-P&S model, the Stat-PCR model provides a deterministic prediction which falls within the same category as the Stat-P&S model.

**Comment:** The Referee suggests producing predictions of rainfall and subsequently coupling with a physically-based runoff model as a more obvious approach to predicting streamflow.

**Reply:** This is a valid comment. However, while the method of precipitation runoff routing is well documented, and certainly applicable to the study area, it is perhaps unnecessary considering the correlation between May-August precipitation and October-January streamflow (Pearson's Correlation Coefficient = 0.80) suggests a strong, direct link exists. As such, predicting precipitation to inform a hydrology model is unlikely to add additional (appreciable) skill, while perhaps introducing additional uncertainty. This is further compounded by the relative lack of spatially diverse observational data and the complex topography of the upper basin. Previous research has also found a strong link between precipitation and streamflow within North Central and Central Chile (Waylen and Caviedes 1990; Verbist et al. 2010). Further, a concurrent study (performed by others) utilized the Water Evaluation and Planning (WEAP) model to address the contribution of rainfall runoff to streamflow in the Elqui basin, and found similar skill in predicting October-January streamflow.

**Comment:** The (Giorgi 1990) reference should be updated.

In the working manuscript, we substitute (Giorgi 1990) for (Fowler and Ekström 2009; Rauscher et al. 2010; Kendon et al. 2014) which each cite the use of general circulation models or regional climate models in seasonal and sub-seasonal precipitation forecasting at or below 20 kilometer resolution, as opposed to 600 kilometers.

**Comment:** Given that October-January streamflow is heavily influenced by concurrent season snow-melt, snow cover and snow depth should provide predictive strength.

The link between snow-melt and streamflow in the basins of North Central Chile is well documented (Souvignet et al. 2008; Vicuña, Garreaud, and McPhee 2011; Ribeiro et al. 2015). However, snow depth, snow cover, snow water equivalent (SWE) are not well sampled in the Elqui both spatially and temporally. The Dirección General de Aguas (DGA), the body charged with hydrologic and meteorological monitoring for Chile, provides SWE for a single location (La Laguna) for the period 1976-2005, which includes significant data gaps. The correlation between May-August SWE and October-January streamflow (Pearson's Correlation Coefficient = 0.67) is not as strong as the correlation between May-August precipitation and October-January streamflow (see above), and arguably provides the same information to the model. As such, we retain precipitation observations as a predictor in the Stat-PCR model. We agree with the Referee that snow observations would seemingly be an obvious predictor of streamflow, and have explored this thoroughly, however in this case, for the reasons mentioned above, it explains less of the overall variance in streamflow as compared with precipitation. We discuss this explicitly in the manuscript and have further highlighted this point in the working version.

**Comment:** The allocation model description is unclear. Specifically, the end of year (February) target volume seems too high, and the requirement to carry storage shortfall to the next year implies the reservoir has does not replenish. Is this realistic?

While we agree, for a variety of reasons, that a static target volume is generally a suboptimal operational policy, the target volume (50% of maximum storage) is the operating rule enacted by Puclaro's operators as a response to critically low reservoir levels (< 20 MCM) observed during the recent extended hydrologic drought (2009-2014). It was not the purpose of this research to address the performance of existing reservoir operating policies. Rather, we evaluate the performance of the October-January streamflow and climatological forecasts, translated to per-water right allocation values using the reservoir allocation model, against perfect foresight as a means of assessing the value of the forecast. A concurrent project is aimed at optimizing Puclaro's operational policies.

The purpose of carrying the storage shortfall or surplus from February and using it as a constraint or benefit for the subsequent October-January per-water right allocation value is that it recognizes the storage target as non-binding (can be violated by over or under allocation in the previous year), but consequential, in the allocation model. As such, the reasonable place to impose the effect is in terms of the following year's allocation. Effectively, it represents a mechanism for the reservoir operator to compensate for over or under allocation in the previous year. In addition, reservoir replenishment from snow melt typically does not occur until December, which is three months after the allocation issuance date of September 1st, and such is the reason why a forecast is produced. The use of the deficit or surplus becomes a hedge against the uncertainty of the forecast. Ultimately, we believe the carryover of deficit or surplus is an appropriate way to include the operational goal of the reservoir operators.

**Comment:** The summary and discussion are uninformative. As such, they should be split into distinct "Discussion" and "Conclusions" sections and significant effort applied to drawing more scientific conclusions from the research.

We agree with the Referee's comment to break the "Summary and Discussion" section into and "Discussion" and "Conclusions" sections, and have done this in the working manuscript. The Discussion section now more thoroughly describes where models are both successful and limited in terms of prediction, and how the limitations (e.g. tradeoff between lead and skill) of the models we construct align with previous research. The Conclusions section establishes the broader insights gained from the research, including the potential for improved water right allocation efficiency achieved by coupling hydroclimate streamflow prediction with a reservoir allocation framework which may benefit both reservoir operators and water rights holders. In addition, the Conclusion presents the coupled Stat-P&S and Stat-PCR models as achieving both increase in forecast lead while maintaining skill, by adjusting the type of forecast provided (categorical to deterministic). The broader insight gained here is that by sacrificing forecast precision, the lead can be skillfully extended. We hypothesize this information to be of potential value to water rights holders who must make decisions (e.g. cropping) prior to the annual setting of the per-water right allocation value.

**Comment:** The manuscript should be shortened. In depth descriptions of well understood methods and metrics may be removed.

We recognize that the interdisciplinary nature of this manuscript may draw readers who have limited methodological knowledge of hydroclimate prediction and reservoir allocation forecasts. Therefore, we provide explicit detail of both methods and metrics used to construct and evaluate models, respectively. The comment is reasonable, and in the working manuscript we have removed all but necessary discussion of principal component analysis, multiple linear regression, cross-validation, and metrics.

---

## Author Response (AR2)

**REPLIES TO COMMENTS**

**Reply to the Editor**

5 We thank Referees 1 & 2 for reviewing our manuscript and providing constructive, actionable feedback. In compliance with the request of the Editor, below we provide our responses to concerns raised by Referee 2 during the 'Revision' stage of peer review.

**Comment:** The manuscript has certainly improved since the first version. However, I think that
10 it is still much longer than necessary, and that one can expect the average HESS reader to be familiar with most if not all the presented concepts (and surely be able to look them up in other sources). From that perspective, I expect that a more condensed presentation would make the paper more accessible and increase its impact significantly.

15 **Reply:** We thank Referee 2 for acknowledging improvements made to the manuscript. We have made a significant effort to balance (decrease) the length of data, methods, and metrics discussion while maintaining contextual details for readability and reproducibility. Further, many of the additions to the manuscript due to comments by Referee 1, which we agree are necessary for completeness, are in sections which Referee 2 suggests consolidation. As such,
20 we feel strongly that the level of detail provided in the manuscript is appropriate. Still, we have further condensed some explanatory language in section 2 "Modelling Framework and Performance Metrics".

**Comment:** Similarly, the structure has improved, but especially the modelling section still
25 combines introduction with methodology and process description in an uncomfortable way. A more traditional (and condensed) structure in which the model justification is followed by a concise description of the technical details of the model implementation would in my opinion help both readability and reproducibility.

30 **Reply:** Again, we thank Referee 2 for noting improvements to the manuscript. Our position is that completely separating data from the three modelling approaches is unnecessary and would likely *add* to manuscript length, without significant gains to readability and reproducibility. Each model uses unique variables, data sets, data time sequences, or configurations which we assert are best presented as a precursor to each model. We recognize this approach to
35 manuscript presentation is less conventional than traditional approaches. To alleviate this concern, we add subheadings 2.1.1 and 2.2.1 "Data and Predictor Selection", 2.1.2 "Statistical Modelling Approaches", and 2.2.2 "Dynamic Model Informed Statistical Modelling Approach" as a means of segregating each model's data from its approach within the same section heading.

40

**LIST OF RELEVANT CHANGES MADE TO MANUSCRIPT**

1. P7, L135 Add sub-heading 2.1.1 Data and Predictor Selection

2. P11, L187 Add sub-heading 2.1.2 Statistical Modelling Approaches

3. P11, L193-199 Various deletions and rewording

4. P112, L207-209 Delete "The Stat-P&S approach utilizes Niño 3.4 Index values, prior to ONDJ season of interest, to provide a categorical streamflow prediction." This language is potentially unnecessary.

5. P12, L222 Add subheading 2.2.1 Data and Predictor Selection

6. P12, L 225-226 Delete "GCMs have proven skilful in prediction of large scale physical processes such as SSTs and pressure systems, however, their" and retain the following sentence which focuses on GCM limitations in prediction of smaller scale climate variables, e.g. precipitation.

7. P13, L239 Add sub-heading 2.2.2 Dynamic Model Informed Statistical Modelling Approach

8. P16, L302-303 Delete "and is the count of years predicted correctly in a category, divided by the number of years observed in the same category." This language is unnecessary as Hit Score calculation is likely well-understood by HESS readers.

[revised manuscript text omitted]

---

## Author Response (AR3)

**REPLIES TO COMMENTS**

**Reply to the Editor**

We thank the Editor for reviewing our manuscript and providing constructive, actionable feedback. Below we provide our responses to concerns raised during the 'Minor Revision' stage.

**Comment:** As you can see from the reviewer's response, one of them still thinks that the paper would benefit from condensing it and restructuring it a bit. I would recommend that you shorten it further, especially the abstract, and attend to some other minor corrections. I will then deem this paper ready for submission.

**Reply:**

We have further reduced the length of the manuscript (~1 page less text) in response to Editor and Reviewer #2 comments.

1) We have addressed the abstract length, condensing by 25%, while retaining pertinent methodological and results detail.
2) Changes to the structure of the data and methods section have been made to more concisely and neatly present the data utilized for the statistical streamflow forecast models.

The minor corrections, including date and unit discrepancies, have been made compliant with HESS standards.

**LIST OF RELEVANT CHANGES MADE TO MANUSCRIPT (Refer to "Manuscript_HESS_markedup")**

1. P1, L8-25: Abstract consolidation.
2. P8-13, L157-244: Considerable reorganization of data and methods presentation. Specifically, the language pertaining to the collection and post-processing of data is extracted from the description of the statistical modelling approaches and presented in a more concise way in Section 2.1.1.
3. P12, L227-235: Removal of basic explanatory language pertaining to principal component regression.

[revised manuscript text omitted]